# Genome-wide promoter responses to CRISPR perturbations of regulators reveal regulatory networks in *Escherichia coli*

Yichao Han [1], Wanji Li[1], Alden Filko[1], Jingyao Li [1] & Fuzhong Zhang [1,2,3] ✉

Elucidating genome-scale regulatory networks requires a comprehensive collection of gene expression profiles, yet measuring gene expression responses for every transcription factor (TF)-gene pair in living prokaryotic cells remains challenging. Here, we develop pooled promoter responses to TF perturbation sequencing (PPTP-seq) via CRISPR interference to address this challenge. Using PPTP-seq, we systematically measure the activity of 1372 *Escherichia coli* promoters under single knockdown of 183 TF genes, illustrating more than 200,000 possible TF-gene responses in one experiment. We perform PPTP-seq for *E. coli* growing in three different media. The PPTP-seq data reveal robust steady-state promoter activities under most single TF knockdown conditions. PPTP-seq also enables identifications of, to the best of our knowledge, previously unknown TF autoregulatory responses and complex transcriptional control on one-carbon metabolism. We further find context-dependent promoter regulation by multiple TFs whose relative binding strengths determined promoter activities. Additionally, PPTP-seq reveals different promoter responses in different growth media, suggesting condition-specific gene regulation. Overall, PPTP-seq provides a powerful method to examine genome-wide transcriptional regulatory networks and can be potentially expanded to reveal gene expression responses to other genetic elements.

Information about the bacterial cellular response is often encoded in promoters and affected by transcription factors (TFs), which control both the timing and level of gene expression. Characterizing the transcriptional regulatory network (TRN) between TFs and promoters is essential for functional genomics, systems biology, and genetic engineering applications. The genome-scale TRN contains massive amounts of information: *Escherichia coli*, for example, possesses at least 183 confirmed TF-encoding genes and 2619 operons according to RegulonDB 10.0[1], corresponding to ~500,000 (183 × 2619) possible TF-operon responses. RNA-seq and microarrays are the most common methods for exploring genome-scale transcriptomic responses to a perturbed TF activity, but identifying responsive genes for all TFs

would require hundreds of RNA-seq or microarray experiments, consuming excessive resources and time[2–9]. Recent advances in single-cell RNA-seq and CRISPR-based perturbations allowed systematic analysis of transcriptional response to various genetic perturbations in eukaryotes[10–12]. However, these methods have not been able to illustrate prokaryotic TRNs at whole genome scales due to the low coverage of bacterial single-cell RNA-seq (less than 10%)[13]. Genome-wide promoter mutational scanning presents another high-throughput method for identifying cis-regulatory elements (CREs) on promoters[14–18]. While powerful, this method alone, without prior knowledge of TF binding sites, cannot provide information about which TF a promoter can respond to. Moreover, when multiple TFs

[1]Department of Energy, Environmental and Chemical Engineering, Washington University in St. Louis, Saint Louis, Missouri, USA. [2]Division of Biological and Biomedical Sciences, Washington University in St. Louis, Saint Louis, Missouri, USA. [3]Institute of Materials Science and Engineering, Washington University in St. Louis, Saint Louis, Missouri, USA. ✉e-mail: fzhang@seas.wustl.edu

bind to the same location, mutational scanning data alone cannot quantify the effect of each TF.

To overcome these limitations, we develop a massively parallel method to measure genome-wide promoter activities in response to CRISPR interference (CRISPRi)-based TF knockdown (TFKD). This method called Pooled Promoter responses to TF Perturbations via sequencing (hereafter PPTP-seq), allows us to examine the regulatory effects in living cells of hundreds of TFs and thousands of promoters of a bacterial genome, all with a single assay lasting just two weeks. Further, PPTP-seq produces homogeneous data for evaluating all regulatory responses under identical growth conditions, avoiding extensive normalization steps in data processing. We apply PPTP-seq to study the *E. coli* TRN in three different grow media (minimal glucose, minimal glycerol, and LB media) and obtain the most comprehensive TF-promoter activity profiles so far. Our study uncovers multiple regulatory responses, including TF autoregulatory responses, complex transcriptional control of metabolic pathways, promoter responses to co-regulation from multiple TFs, and condition-specific gene regulation.

## Results

### PPTP-seq development and validation

PPTP-seq uses plasmid to integrate each CRISPRi-based TF perturbation and each promoter activity reporter into one construct. Each plasmid contains a CRISPRi cassette that constitutively expresses a single guide RNA (sgRNA) to repress a specific TF in the genome[19] and a promoter-reporter cassette to measure the activity of a specific promoter under the TF-repressed condition (Fig. 1a, b). A self-cleaving ribozyme, RiboJ, was inserted between the promoter and the *gfp* reporter gene to produce invariant mRNA sequences, thus eliminating the interference of different promoter sequences with *gfp* mRNA stability[20].

To profile genome-wide transcriptional responses for all TFs in *E. coli*, we constructed a combinatorial plasmid library consisting of both a sgRNA library and a promoter library (Fig. 1c). The sgRNA library contains 183 TF-targeting sgRNAs that repress every single known TF gene in the *E. coli* genome (Supplementary Data 1), and contains five

non-targeting sgRNAs as negative controls. The promoter library contains 1372 native promoters that cover more than 50% of all operons in *E. coli*[21] (Supplementary Data 2). The combinatorial plasmid library was transformed into *E. coli* strain FR-E01, which carries a *dCas9* gene in its chromosome. Transformed cells were first grown in minimal glucose medium to a steady state and sorted into 16 bins based on their fluorescence intensity (Supplementary Fig. 1a). More than 20 million cells (including all 16 bins) were sorted in each replicate (Supplementary Fig. 1b and Supplementary Data 3), and their plasmids were sequenced using the NovaSeq S4 XP Platform, generating an average of 420 million reads from each replicate (Supplementary Fig. 1c and Supplementary Data 3). To estimate promoter activities under each perturbed TF condition, sequencing read counts across the bins were first converted to cell count distribution for each individual variant, followed by fitting into log-normal distribution by maximum-likelihood estimation[22–24] (Supplementary Fig. 2 and "Methods").

Measured promoter activities were highly consistent between independent biological replicates performed in different weeks, with replicate correlation ranging between 0.90 and 0.95 (Supplementary Fig. 3a). Across three independent replicates, the promoter activities of 201,433 library members (i.e., 201,433 different TF-promoter pairs, 81% of the entire library) passed our quality filters (Supplementary Fig. 3b, "Methods"). For most promoters, the median activity of a promoter across all TFKD conditions was consistent with its activity in negative controls (Fig. 1d and Supplementary Fig. 4). We found that more than 98% of TF-promoter pairs fell within the two-fold-change boundaries of the median activity, indicating robust promoter activities in most TFKD conditions[18,25].

CRISPRi can impair cell growth if essential genes are targeted. Seven TF-targeting sgRNAs (*alaS*, *bluR*, *dicA*, *dnaA*, *iscR*, *mraZ*, and *nrdR*) had substantially reduced reads (fewer than 10,000 reads per sgRNA compared to an average of 4.8 million reads per sgRNA). Among them, *alaS*, *dicA*, and *dnaA* are essential genes whose deletion led to cell death[26,27]. CRISPRi polarity[28,29] can also lead to the repression of essential genes that are located downstream of a targeting TF within the same operon. This explains the substantially reduced reads for *iscR*, *mraZ*, and *nrdR*.

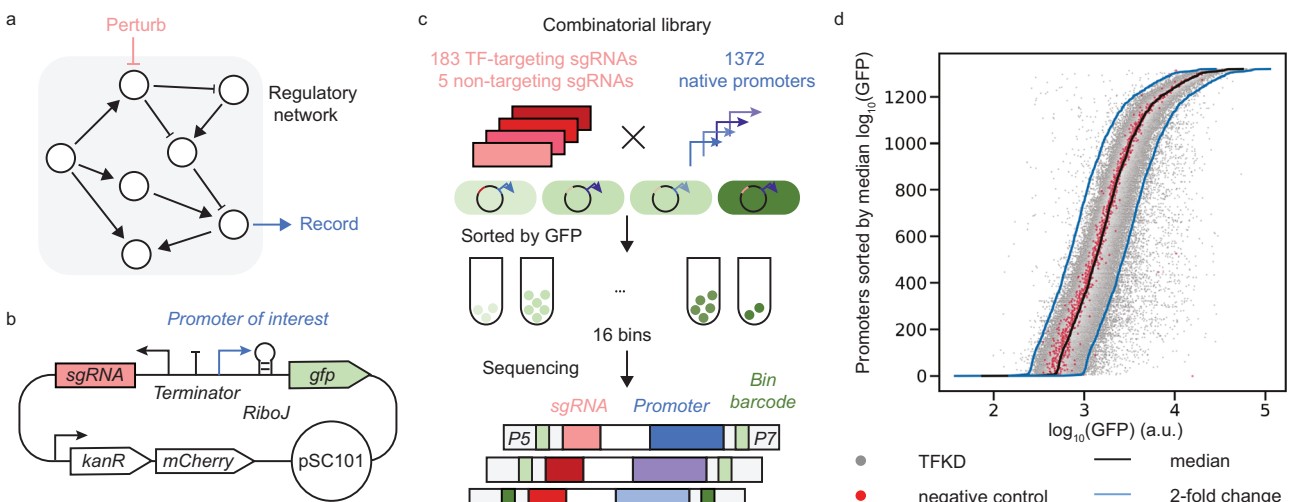

**Fig. 1 | Genome-wide promoter activity profiles of TFKD measured by PPTP-seq. a** Schematic of a regulatory network. Perturbing regulators and the recorded responses of genes are used to infer regulatory interactions. **b** Reporter plasmids used to quantify promoter activity under CRISPRi-based regulator perturbation. A native promoter was cloned upstream of the *gfp* gene, and a sgRNA was inserted downstream of a constitutive promoter. **c** Massively parallel promoter activity measurements for a combinatory library. A combinatory library of more than $2.5 \times 10^5$ sgRNA-promoter pairs was sorted into 16 bins according to their GFP expression levels. The sgRNA and promoter regions in each bin were sequenced to estimate perturbed promoter activity for each sgRNA-promoter pair. **d** Sorted promoter activities of all promoters. The gray and red dots respectively represent promoter activities in strains with TF-targeting sgRNAs and negative control sgRNAs. The black line represents sorted median promoter activities across all TFKD conditions. The blue lines indicate 2-fold changes from the median activities. a.u. arbitrary units. Source data are provided as a Source Data file.

We further evaluated the CRISPRi repression efficiency using both TF's promoter activity measured from PPTP-seq (Supplementary Fig. 5a) and transcript level measured from RT-qPCR (Supplementary Fig. 5b). The two methods respectively found 95% and 86% of tested TFs showed significant repression (Student's *t*-test *P*-value < 0.05) compared to their corresponding controls containing non-targeting sgRNAs (Supplementary Note 1). We further found a clear negative correlation between the degree of CRISPRi repression and TF expression level measured from TF's promoter activity (Supplementary Fig. 5c, d). This explains the lack of repression for the small fraction of TFs (e.g., *qseB* and *ttdR*).

To further validate the promoter activities measured by PPTP-seq, we randomly selected five promoters, which involve a diverse range of gene functions. We then individually measured their activities in response to CRISPRi repression of nine representative TFs (and one non-targeting sgRNA as a negative control), using a plate-reader-based whole-cell fluorescence assay (Supplementary Fig. 6a). Of these 50 sgRNA-promoter pairs, 45 were quantified by PPTP-seq and were highly consistent with individual whole-cell fluorescence measurements (Supplementary Fig. 6b, Pearson's *r* = 0.95), confirming the high quality of our pooled measurements. The other five combinations were missing in all three replicates due to their low read counts. This small dataset also contained the regulatory effects of five known

direct interactions and one indirect interaction in RegulonDB[1] (Supplementary Fig. 6c).

We also compared our promoter activity measurements to previously published datasets from other independent experiments. Promoter activities measured from PPTP-seq (using the negative control strains) correlated with transcript levels measured from RNA-seq[30] and promoter activities individually measured using flow cytometry[31] (Supplementary Fig. 7a–c, Pearson's *r* = 0.68 and 0.74, respectively). Additionally, fold change in promoter activity upon TFKD measured from PPTP-seq is also qualitatively consistent with that measured from EcoMAC microarray[32] for a few known regulatory interactions in RegulonDB[1] (Pearson's *r* = 0.51, Supplementary Fig. 7d).

### Genome-wide TF-dependent promoter responses

We quantified promoter activity changes by TFKD relative to negative controls (Supplementary Fig. 4) and modeled the replicated data as log-normal distributed to determine statistical significance. From the 201,433 measured promoter activities, single TFKDs led to upregulation in 3720 TF-promoter pairs and downregulation in 338 pairs (>1.7-fold in promoter activity, *q* < 0.01; Fig. 2a) in minimal glucose medium. Most TFs regulate fewer than ten promoters, while a few TFs affect more than 100 promoters (Fig. 2b). We also found promoters that are regulated by multiple activators (leading to downregulation by

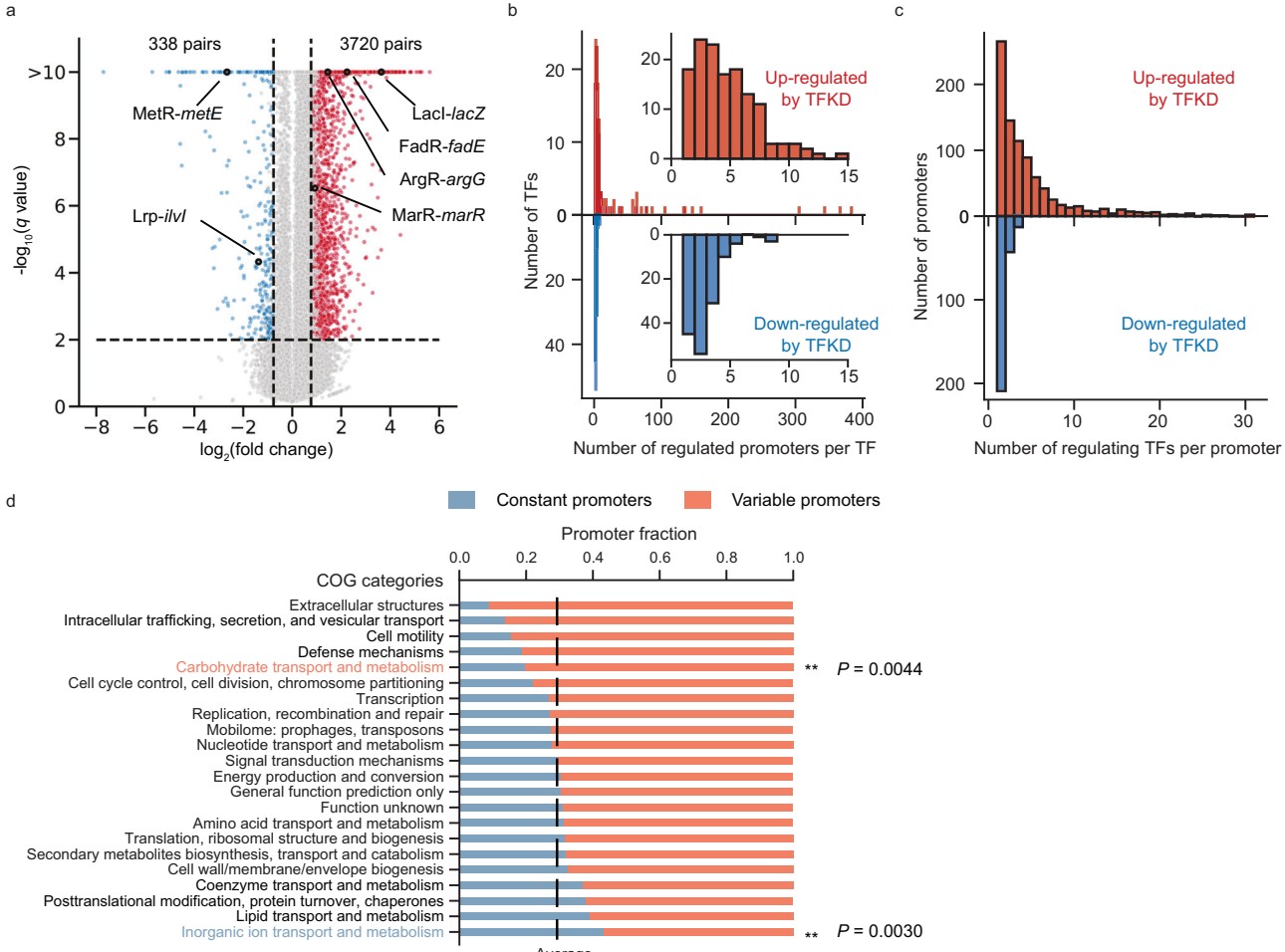

**Fig. 2 | Genome-wide promoter responses to TFKD in *E. coli*. a** Promoter activity changes by TFKD. Dashed lines indicate cutoffs for statistically significant (*q* < 0.01) and substantial (>1.7-fold change) effects. Each dot represents a TF-promoter pair. Upregulation and downregulation by TFKD are shown in red and blue, respectively. A few known interacting TF-promoter pairs are labeled. **b** Histogram of the number of regulated promoters per TF. Inset in (**b**) shows histograms over a smaller range. **c** Histogram of the number of regulating TFs per promoter. **d** Fractions of constant promoters and variable promoters in each COG category. All COG categories of genes in an operon controlled by a promoter are assigned to the promoter. The dashed line indicates the average fraction of constant promoters over all COG categories. Statistical significance is determined by one-sided Fisher's exact test. **\*\****P* < 0.01. Source data are provided as a Source Data file.

TFKD in Fig. 2c) are much less abundant than those regulated by multiple repressors (leading to upregulation in Fig. 2c). The most common regulatory effect on a regulated promoter observed in PPTP-seq was single regulation by a single activator or a single repressor (30%, Fig. 2c and Supplementary Fig. 4), which was consistent with previous datasets measured using other methods[1,14].

Collectively, we identified 936 (71% of 1323 measured promoters) variable promoters with significant activity change under at least one TFKD condition (Supplementary Note 2), and the other 29% of the promoters were considered as constant promoters. Clusters of Orthologous Genes (COG) analysis[33] of all downstream genes of these promoters indicated that genes expressed by variable promoters are enriched in the COG class of "Carbohydrate transport and metabolism" ($P = 4.4 \times 10^{-3}$) (Fig. 2d), specifically KEGG pathways in galactose metabolism (eco00052), pentose and glucuronate interconversions (eco00040), starch and sucrose metabolism (eco00500), and amino sugar and nucleotide sugar metabolism (eco00520). Variable promoters also control genes in flagellar and pilus (Supplementary Data 4). The results suggested that these functions or activities are more readily subject to regulation under different condition changes. Genes expressed by constant promoters are enriched in "inorganic ion transport and metabolism" ($P = 2.6 \times 10^{-3}$), specifically sulfur metabolism (eco00920), ion transport (GO:0006811), and iron ion homeostasis (GO:0055072) (Supplementary Data 4), suggesting that these genes play housekeeping roles (Fig. 2d).

## TF promoter response to perturbation

We systematically investigated whether a TF's promoter can be affected by itself or other TFs. A perturbation-response network between TFs was constructed, where activation and repression represent down- and upregulation by CRISPRi knockdown of an upstream TF, respectively (Fig. 3a). In minimal glucose medium, a total of 26 activations and 339 repressions were observed between 126 TFs (Supplementary Data 5). Within this dataset, no mutual regulation or repressilators of three or more TFs were observed, likely due to low expression or missing allosteric regulation for some TFs when cells are growing in minimal glucose medium (Supplementary Note 3).

We then examined TF autoregulatory responses, which have been challenging to study using other methods due to the coupling between perturbation and readout. We identified 12 autoregulated TFs with strong perturbation effects (>1.7-fold in promoter activity, $q < 0.01$) in minimal glucose medium, including two autoregulatory interactions, PgrR and ComR, not present in RegulonDB (Fig. 3b). Meanwhile, several previously identified autoregulated TFs (e.g., PhoB, Fur, LldR, etc.) showed only weak perturbation effects (i.e., less than 30% promoter activity change) under our growth conditions in minimal glucose medium. To further validate these findings, we selected seven TF genes and measured their promoter activities across a wide range of TF concentrations using a tunable *E. coli* TF library[34], in which each endogenous TF is replaced by an inducible TF-mCherry fusion (Supplementary Fig. 8). Both *pgrR* and *comR* promoters showed higher activity at lower TF levels, confirming their negative autoregulation. PgrR autoregulation is consistent with the identified PgrR binding site on its promoter region[35]. Except for ZraR, four out of five previously identified autoregulated TFs displayed negligible promoter activity changes over a wide TF level range. Thus, the results from the tunable TF library were mostly consistent with PPTP-seq. Our results also suggest that some previously identified TFs lack autoregulatory response when cells are growing in minimal glucose medium and may occur under other growth conditions[36-39], so the interpretation of TF regulation should consider the condition dependency.

## Transcriptional regulation of one-carbon metabolism

PPTP-seq data also allows us to systematically examine gene regulation on complex metabolic pathways. As an example, we selected the one-carbon metabolism (OCM), in which transcriptional regulation was not well characterized in bacteria. OCM is tightly associated with the synthesis of nucleotides, amino acids, and two essential cofactors——tetrahydrofolate (THF) and S-adenosylmethionine (SAM), and it plays important roles in cell survival and growth. However, due to the presence of multiple metabolic cycles and interconnected pathway structures, dissecting the regulatory function of OCM remains challenging.

We identified 28 TF genes that can affect at least one promoter in OCM (Supplementary Fig. 9). A few genes in methionine and SAM biosynthesis, such as *metA*, *metE*, and *metK*, were observed to be upregulated by *metJ* knockdown, recapitulating the known feedback control of SAM biosynthesis via MetJ[5,40] (Fig. 4a). Additionally, we found that *metA*, *metE*, and *metK* were also regulated by other TFs, but in distinct patterns (Fig. 4b). For example, *metE* was found to be activated only by *metJ* knockdown, while *metK* was upregulated by knockdown of ten different TFs. This finding is intuitively surprising because MetE and MetK catalyze two consecutive reactions in the methionine cycle, and enzymes from the same pathway are often co-regulated[41]. The different regulations on *metE* and *metK* thus indicate that enzymes catalyzing consecutive steps can have distinct cellular functions: MetE synthesizes methionine for protein synthesis, and MetK produces SAM as a cofactor for metabolic reactions (Fig. 4a).

The PPTP-seq dataset also revealed the regulatory functions of MetR, previously known only as a regulator of methionine biosynthesis. We found that *metR* knockdown affected multiple genes in the folate cycle and folate biosynthesis (e.g., *metF*, *thyA*, and *folE*; Fig. 4a), not present in RegulonDB[1]. Previous DAP-seq binding analysis using purified TFs and genomic DNA fragments identified MetR binding sites at *metF* and *folE* promoters[42], but the in vivo regulatory responses have never been tested. We further verified these regulatory responses using a MetR knockdown strain from the tunable TF library[34] (Fig. 4c). These findings allow us to discover metabolic feedback control mechanisms in *E. coli* OCM under homocysteine-starved conditions because MetR binding to DNA requires homocysteine activation[43]. When homocysteine is limited, cells cannot produce sufficient methionine for translation initiation and elongation. To quickly rescue the cells from their methionine-limited state, MetR-repression of *metF* must be alleviated, increasing the amount of 5-methyl-THF and preparing for rapid methionine synthesis when the homocysteine level is sufficiently restored. Meanwhile, upregulated *metF* and *thyA* by MetR also increase 5,10-methylene THF consumption, which simultaneously reduces 10-formyl-THF due to reversible reactions between these THF species (Fig. 4a). Low 10-formyl-THF and methionine can further result in the insufficient formation of initiator tRNA to slow down translation. Additionally, we found that MetR activates *folE*, whose enzyme product catalyzes the first step in folate biosynthesis (Fig. 4a). Thus, homocysteine limitation can also repress *folE*, thereby decreasing folate biosynthesis. Taken together, these phenomena suggest that MetR helps to block protein translation initiation and folate synthesis in response to low homocysteine and accumulates 5-methyl THF to prepare for rapid methionine biosynthesis once homocysteine is available.

## Strongly bound rather than weakly bound TFs tend to affect promoter activity

Our genome-wide promoter activity measurements from perturbed TF levels can provide information that complements TF-promoter binding datasets from ChIP-seq, ChIP-exo, DAP-seq, gSELEX, and curated TF binding sites (TFBSs) in RegulonDB[1,42,44,45], yielding knowledge about direct and functional TF-promoter interactions. In total, out of the 4058 regulatory responses identified by PPTP-seq in minimal glucose medium, 225 have binding evidence from DAP-seq, and an additional 256 have binding evidence from other binding datasets,

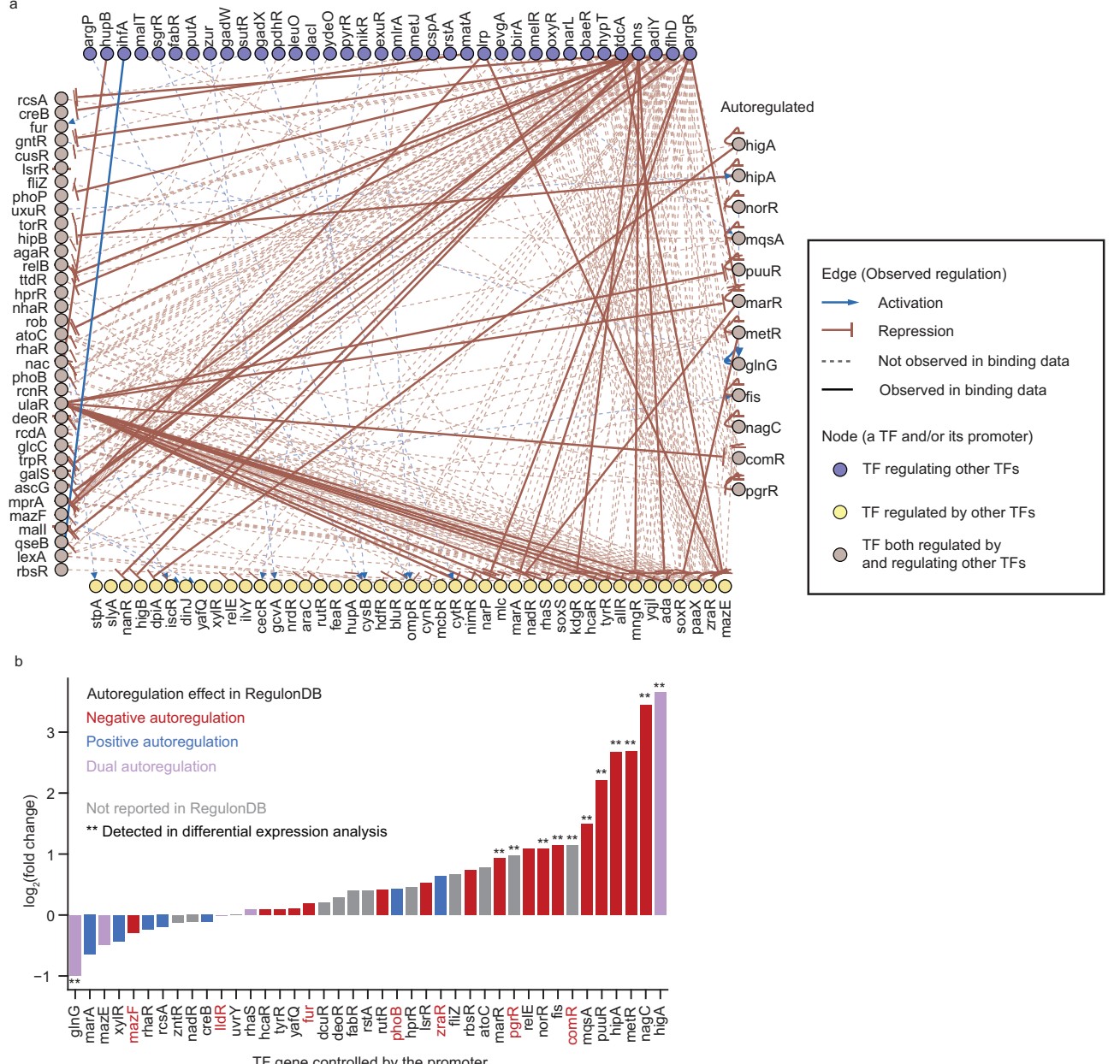

**Fig. 3 | Perturbation effects of transcription factors on their promoters.**
**a** Perturbation-response network of TFs constructed using PPTP-seq data in minimal glucose medium. **b** Autoregulation of TFs identified by PPTP-seq in minimal glucose medium. Promoter activity fold changes upon the knockdown of TF controlled by the promoter. TF gene names marked in red were selected for validation. Source data are provided in Supplementary Data 5.

altogether representing 12% (481/4058) of the PPTP-seq identified responses (Fig. 5a, b, Supplementary Data 6). For 127 TFs with binding site information, on average, 23% of regulated promoters per TF were presumably direct targets (Fig. 5c). For the rest 56 TFs, their TFBSs were either not in our promoter library or not identified yet. Among the 481 regulatory responses with binding evidence, only 78 of them were found in the TF-operon network in RegulonDB, and the rest 403 TF-promoter responses may contribute to regulatory interactions not present in RegulonDB in minimal glucose medium (Supplementary Table 1).

In general, PPTP-seq results and the binding datasets have a small overlap in TF-promoter interaction pairs (Fig. 5a), which is consistent with the low overlaps between similar comparisons on specific TFs (GadX, GadW, Fur, and SoxS) in *E. coli*[36,46,47] and between eukaryotic transcriptional response and TF binding datasets[3,48]. This can be

caused by low TF expression levels, low TF activity (affected by other molecules), and/or complex regulatory patterns. We individually examined two promoters that have multiple different TF binding sites (Supplementary Note 4 and Supplementary Fig. 10). We found the lack of response can be explained by the context-dependent transcriptional regulation[49]——regulatory function of one TF affected by other TFs bound on the same promoter. Further, we found that deactivating the regulating TF can lead the promoter to respond to previously non-regulatory TFs (Supplementary Note 4 and Supplementary Fig. 10h, i). These observations indicate that TF-promoter binding is not sufficient for response, and *E. coli* uses layered control to achieve complex logic for gene expression. In RegulonDB, 48% of regulated promoters have more than one functional TF binding site (Supplementary Fig. 11), suggesting that such context-dependent transcriptional regulation can be ubiquitous in *E. coli*.

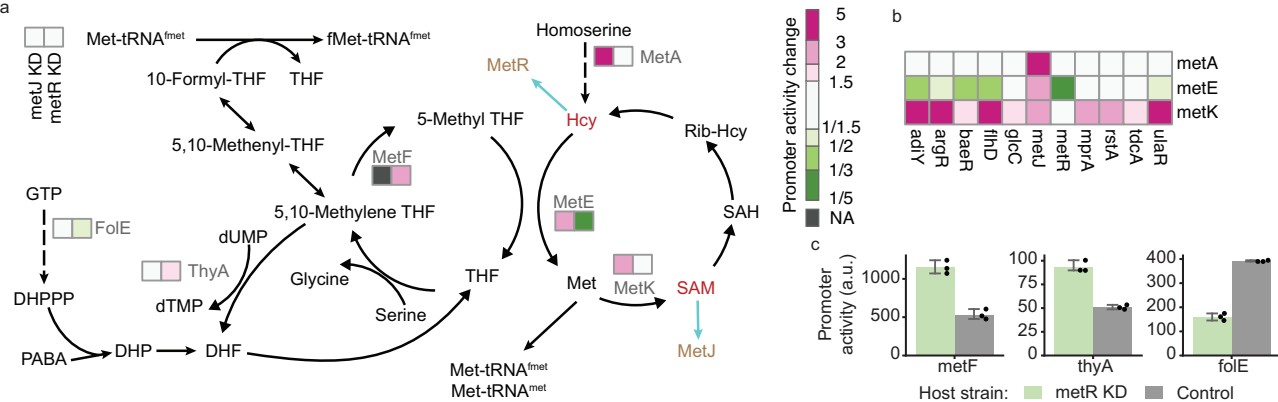

**Fig. 4 | Promoter activity changes in one-carbon metabolism. a** Promoter activity changes in response to *metR* and *metJ* knockdown by CRISPRi. Hcy and SAM control the activity of MetR and MetJ, respectively. NA not applicable, KD knockdown, GTP Guanosine-5′-triphosphate, DHPPP 6-hydroxymethyl-7,8-dihydropterin pyrophosphate, PABA *para*-aminobenzoic acid, DHP dihydropteroate, DHF dihydrofolate, THF tetrahydrofolate, dUMP deoxyuridine monophosphate, dTMP deoxythymidine monophosphate, Met L-methionine, fMet N-formylmethionine,

Hcy L-homocysteine, SAM S-adenosylmethionine, SAH S-adenosylhomocysteine, Rib-Hcy S-ribosyl-L-homocysteine. **b** TF-dependent promoter activity changes for *metA*, *metE*, and *metK*. Each row represents a promoter, and each column stands for a TFKD condition. **c** Validation of MetR targets. Promoter activities were measured in a *metR* knockdown strain and, as a control, in a wild-type *E. coli* strain. Data are presented as means ± SD of three replicates from different days. a.u. arbitrary units. Source data are provided as a Source Data file.

We sought to explore what general features determine whether a potentially bound TF can regulate promoter activity under our experimental condition (i.e., growing in minimal glucose medium). For each TF binding site, we focused on the binding location, TF concentration, and binding strength. We found that binding sites from both regulating and non-regulating TFs were centered around the transcription start site (TSS) of a promoter[50] (Fig. 5d) and that regulating TFs had a significantly higher concentration in cells over non-regulating TFs (Fig. 5e, f). Additionally, previous biophysical models indicate that TF-DNA binding energy can predict fold changes in promoter response[16,51–53]. We first hypothesized that when a TF has binding sites at multiple promoters, it tends to regulate its targets with the strongest binding strength. To test this hypothesis, we normalized the binding strength of each TF-promoter pair to the maximum binding strength for that TF (called "relative binding strength per TF"). On average, the relative binding strength per TF was slightly weaker for regulatory TF-promoter pairs than for non-regulatory TF-promoter pairs (Fig. 5g, h). This unexpected result suggests that TFs do not necessarily regulate their most tightly associated promoters. We then considered the affinity of all TFs binding to the same promoter and normalized the binding strength of each TF-promoter pair to the maximal strength of the most tightly associated TF for each promoter (called "relative binding strength per promoter") (Fig. 5i). Results indicate that for each promoter, TFs with stronger binding are more likely to cause promoter activity change. Taking these findings together, the relative binding strengths of TFs on a promoter are a major determinant of promoter response.

### Condition-specific regulatory networks
To explore genome-scale regulatory networks at conditions other than minimal glucose medium, we further performed PPTP-seq experiments for cells grown in LB and minimal glycerol media. A total of 5279 and 3810 TF-promoter responses were identified in LB and minimal glycerol media, respectively (Supplementary Fig. 12). The larger number of responses seen in LB was partially caused by high TF activity of a few TFs that have specific effectors in rich media (Supplementary Table 2). Comparing these datasets with that collected from minimal glucose medium, 867 TF-promoter pairs appeared in all three conditions, with 1901, 2274, and 3495 pairs appearing only in one condition, suggesting TF-promoter responses are highly condition-specific (Fig. 6a). The upregulated TF-promoter pairs by TFKD (TF repression) have more overlaps among these three conditions than

downregulated pairs (TF activation, Fig. 6a), suggesting that TF activation is more sensitive to growth conditions (e.g., affected by allosteric regulation) than TF repression. We examined a few individual TFs with known targets (Supplementary Data 7) that have distinct regulatory responses in different conditions (Fig. 6b). For example, repression of *lacZ* promoter by CRP was not detected in minimal glucose medium due to low cAMP concentration[54], but was observed in LB medium. Similarly, activation of the maltose transporter *malK* by MalT was observed in LB medium but not in the minimal glucose medium, because expression of *malT* requires CRP activation[55]. On the other hand, activation of *metE* by MetR was observed in minimal glucose and glycerol media but not in LB medium. This is likely caused by repression of *metE* by MetJ at high SAM concentration[56]. Our data show that many regulatory responses are condition-dependent (Fig. 6b) and highlight that growth condition needs to be specified when describing the regulatory network.

## Discussion
In summary, PPTP-seq is a powerful high-throughput method for measuring genome-wide promoter responses to TF perturbations in living cells. This method allows us to interrogate the regulatory functions of 181 *E. coli* TFs in a single assay. RNA-seq is currently the most common technique to study genome-wide TF regulation of living cells. To date, RNA-seq profiles of only 33 *E. coli* TFs were directly assessed[5,8,9], while PPTP-seq increased this number substantially. Further, ChIP-seq was specifically developed for identifying genome-wide TF binding sites in living cells. So far, only 12 *E. coli* TFs have been reported from 28 ChIP-seq databases, while PPTP-seq reports 15-fold more TFs in a single study. Meanwhile, PPTP-seq involves perturbation of TF expression level, similar to methods that perturb TF-promoter binding affinity via mutating CREs[14–16] and methods that perturb TF activity[57]. PPTP-seq identified many regulatory responses that are condition-specific (Fig. 6) and not seen from previous binding assays (e.g., DAP-seq, ChIP-seq, ChIP-exo, gSELEX, Fig. 5, and Supplementary Data 6). Each method has its own advantages and limitations. Complementary use of these methods would help to obtain an unbiased understanding of TF regulation.

Results from this work have also revealed multiple regulatory responses. We identified PgrR and ComR as autoregulated TFs and found that TF autoregulation is condition-dependent. We also discovered complex transcriptional control of OCM, especially the additional roles of MetR in regulating the folate cycle and folate

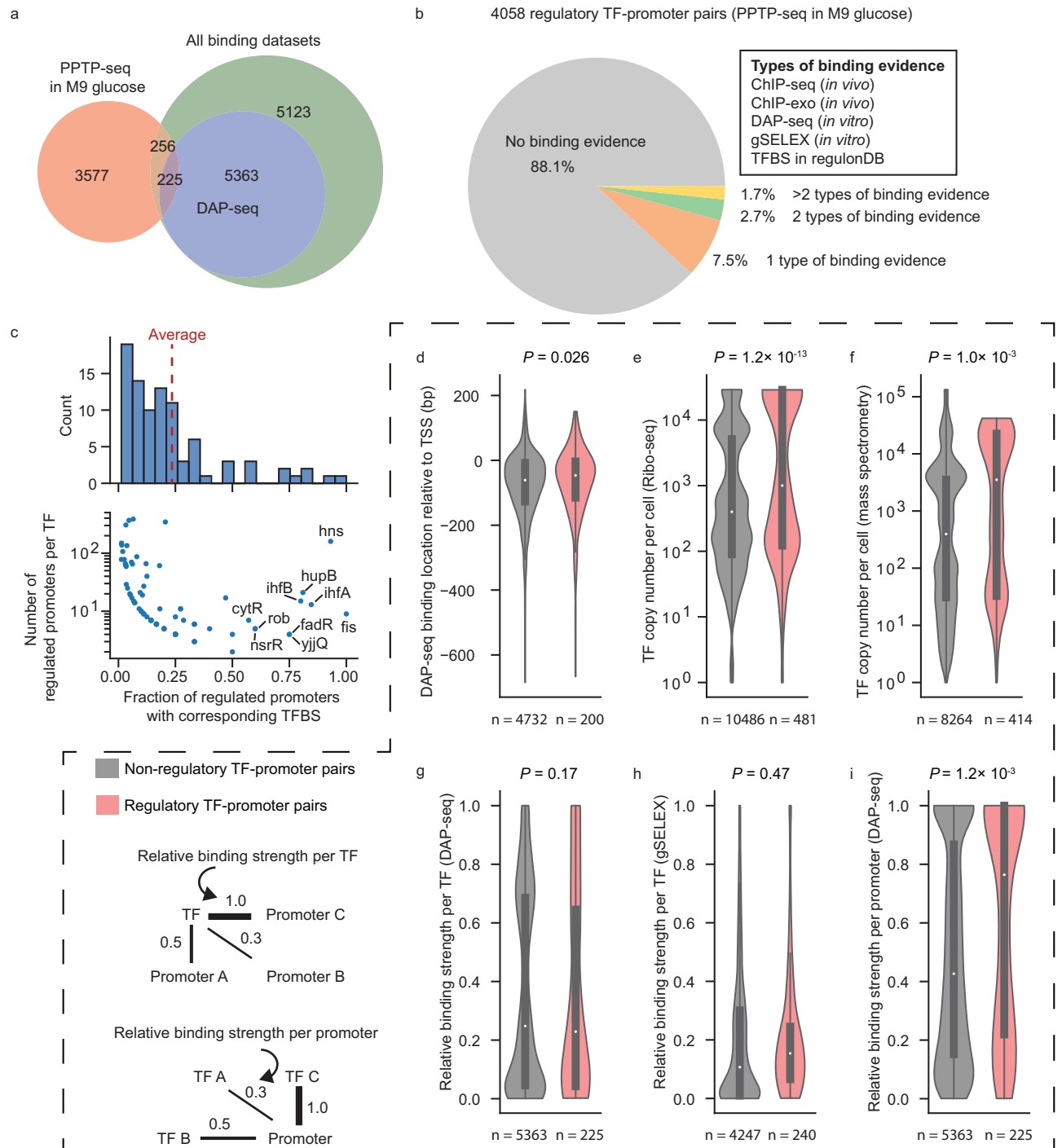

**Fig. 5 | Integrative analysis of promoter response and TF binding. a** Comparison of TF perturbation-response results from PPTP-seq and TF binding results. **b** Fraction of TF-promoter pairs that have binding evidence. **c** Distribution of fraction of regulated promoters with corresponding TFBS for each TF. **d–h** Factors that may affect whether a potentially bound TF on a promoter affects the promoter activity. For each TF-promoter binding interaction, the binding site location in DAP-seq (**d**), TF concentration measured by Ribo-seq (**e**), TF concentration measured by mass spectrometry (**f**), relative binding strength per TF measured by DAP-seq (**g**), relative binding strength per TF measured by gSELEX (**h**), and relative binding strength per promoter measured by DAP-seq (**i**) were considered. The violin plot shows the distribution of data, the central dot in the box represents the median, the box bounds represent the 25th and 75th percentiles, and whiskers represent the minima to maxima values. The number of TFBSs is indicated below. Benjamini–Hochberg adjusted *P*-values were calculated by the Wilcoxon rank sum test. Source data are provided in Supplementary Data 6.

biosynthesis. Although thousands of TF binding sites were identified in *E. coli*, only a small fraction of such interactions cause promoter activity change when perturbing TF expression level. Furthermore, for promoters with multiple TF binding sites, TFs with higher binding affinities are more likely to affect promoter activity than those with lower affinities.

Many regulatory responses identified by PPTP-seq may involve indirect regulatory mechanisms without a binding site identified from the previous datasets. Indirect mechanisms can arise from diverse cellular processes, including regulatory cascade, metabolic state changes, protein-protein interactions, and cell-cycle perturbation[57–60]. Although distinguishing direct versus indirect responses is important

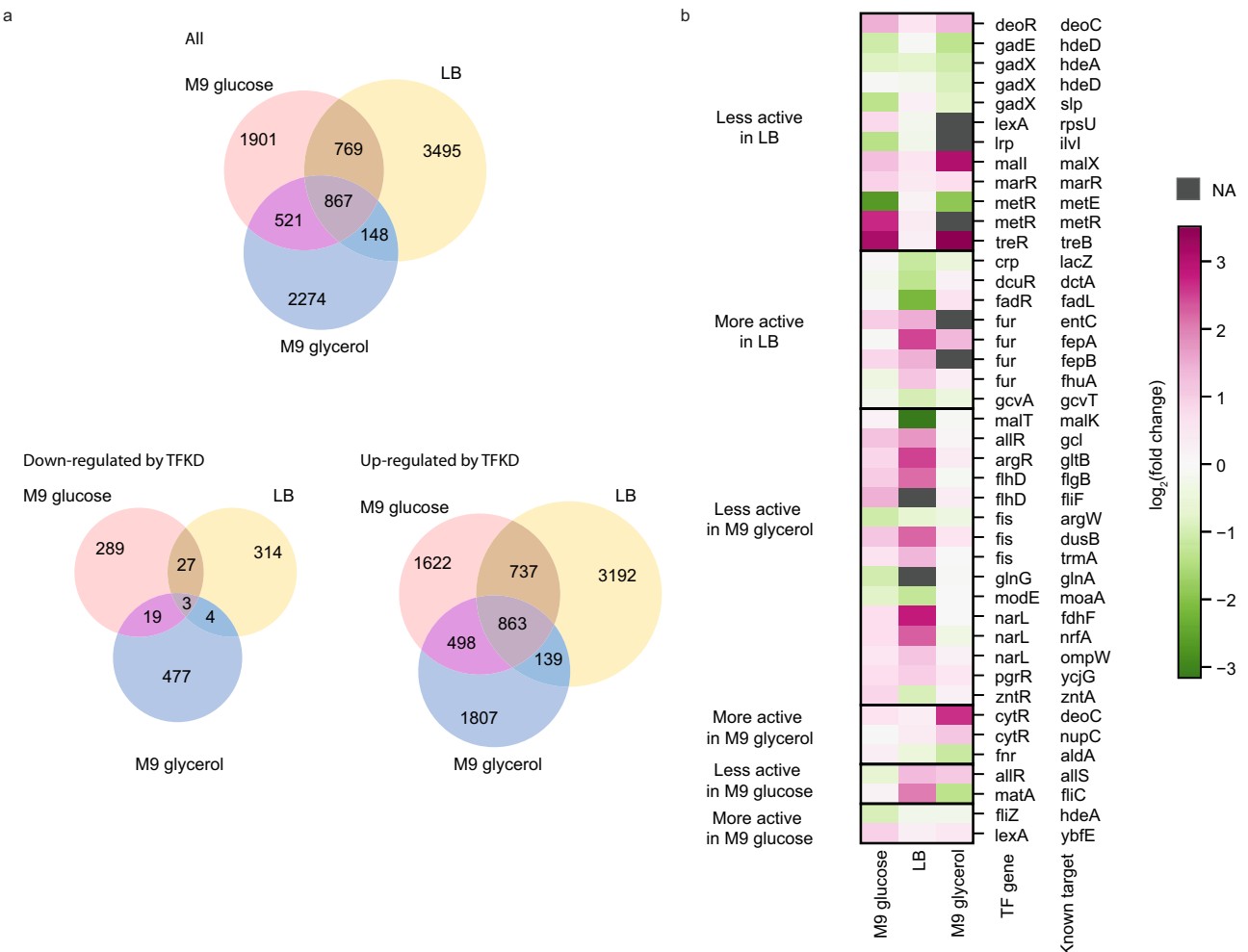

**Fig. 6 | Condition-specific promoter responses to TFKD. a** Comparison of TF perturbation-response results from PPTP-seq at different growth conditions. **b** Known TF-promoter interactions from RegulonDB showed different regulation under different growth media. Source data are provided as a Source Data file.

in understanding network dynamics[61,62] and engineering biosensors[63], our datasets provide genome-wide regulatory phenotypes under different growth conditions and will be useful for a wide range of bio-technology applications, such as engineering dynamic regulation for bio-production[64–66] and identifying new targets for drug development[67,68].

Limitations of PPTP-seq include false positives caused by CRISPRi polarity in bacteria[28,29], where CRISPRi represses genes located in the same operons of a targeting TF. False positives may also result from CRISPRi off-target[69]. Treating CRISPRi polarity and off-target as two independent factors, we expected the false positive rate of PPTP-seq to be lower than 17.8% (Supplementary Note 5). Besides, CRISPRi did not work well for weakly expressed TF genes, leading to false negatives due to insufficient TF repression. Furthermore, PPTP-seq measures expression fold-change from low-copy plasmids that may be smaller than fold-change from single-copy chromosomal promoters. Additionally, some promoters in the library may lack a DNA looping mechanism (e.g., *lacZ*) due to the truncation of additional operator sites located outside the promoter region[70].

PPTP-seq can be applied to other bacterial species because it does not require functional annotations about TF activities and binding motifs. Although this study focused on TFs, PPTP-seq can be modified to explore genome-wide promoter response to other genetic perturbations, such as perturbations of enzymes and transporters, to dissect metabolism-related regulatory networks (Supplementary Fig. 13a). Simultaneous perturbations of multiple TFs using sgRNA

arrays[71] could also be integrated to quantify combinatorial regulatory effects (Supplementary Fig. 13b). Besides genome-scale mapping, PPTP-seq can also explore the regulatory mechanism of complex promoters at a base-pair resolution by perturbing both the binding sites and the expression levels of TFs on these promoters (Supplementary Fig. 13c). We anticipate that PPTP-seq will be a powerful tool for deciphering bacterial regulatory genomes.

## Methods
### Strains, growth media, and DNA libraries
PPTP-seq experiments were performed in *E. coli* strain FR-E01 (Addgene # 118727), a derivative of *E. coli* MG1655 with an aTc-inducible dCas9 expression cassette integrated at the *attB* site of its genome[72]. NEB 10-β competent *E. coli* (New England Biolabs) was used for cloning. Promoter activities were measured in M9 minimal glucose (0.4%) medium (supplemented 1 mM thiamine and 25 ng/μL kanamycin), M9 minimal glycerol (0.5%) medium (supplemented 1 mM thiamine and 25 ng/μL kanamycin), or LB medium (supplemented 25 ng/μL kanamycin). All primer sequences are listed in Supplementary Data 8.

An *E. coli* promoter collection originally constructed by the Alon lab[21] was obtained from Horizon Discovery Ltd. (# PEC3877). All strains in this collection, except for those containing control vectors (pUA139 and pUA66), were grown overnight in LB medium in 96-well deep-well plates. A library was generated by mixing 300 μL of overnight cell cultures from each well using an Eppendorf ep*Motion*® 96 Pipettor.

Pooled plasmids of this collection were extracted from the mixture using a Maxiprep kit (QIAGEN). We noted that "promoter regions" in Alon's collection were defined as entire intergenic regions flanked by about 50–150 bp into the adjacent coding regions[21]; however, many of these "intergenic regions" are located in the middle of an operon and do not contain promoter sequences. These non-promoter regions were excluded from the data analysis.

From the existing TF-gene network (RegulonDB v 10.0), 181 TFs (including heteromultimeric TFs) were identified in *E. coli* that have at least one known target supported by binding of purified proteins or site mutation. Among them, 169 TFs function as monomers or homo-oligomers (encoded by single genes), and 12 TFs function as hetero-dimers or hetero-oligomers (encoded by more than one gene). The 181 TFs are encoded by 183 unique genes; thus, a sgRNA library targeting 183 different TF genes was designed. (Supplementary Data 9). Based on previous CRISPRi screening results for *E. coli*[73], a customized Python script was used to select one sgRNA for each TF gene. Five sgRNAs containing random sequences without off-target candidates in the genome were included as negative controls (Supplementary Data 9). Thus, a total of 188 sgRNA sequences were designed. For each sgRNA, a pair of phosphorylated oligonucleotides were synthesized by IDT and annealed individually. All the oligonucleotide sequences designed for sgRNA cloning are listed in Supplementary Data 9.

## Construction of the pooled combinatorial library

To facilitate the construction of the combinatorial library, we first created the plasmid pYH156, which contains a *gfpmut2* gene under the control of the *lacZ* promoter and a constitutively expressed sgRNA targeting the coding sequence of genomic *lacI* gene using DNA from previously described plasmids[21,74,75]. A self-cleaving ribozyme sequence (RiboJ) was inserted upstream of *gfpmut2* to prevent the untranslated region of different promoters from affecting the mRNA structure. A *mCherry* gene was inserted downstream of the *kanR* gene in the same operon as a control for extrinsic noise. All these genes were cloned on a pSC101 vector backbone (Supplementary Note 6).

The combinatorial library was constructed in two steps. In all cloning steps, Q5 hot-start high-fidelity DNA polymerase (New England Biolabs #M0493L) was used for PCR amplification. The backbone of the plasmid pYH156 was amplified using primers prYH068 and prYH069. In the first step, the vector backbone was first ligated with the sgRNA library by Golden Gate assembly in 96-well PCR plates. The ligation products were pooled and transformed into NEB 10-β competent cells. After growing overnight, more than $10^5$ colonies were scraped from LB agar dishes, followed by plasmid extraction using a Miniprep kit (iNtRON biotechnology), resulting in a plasmid library that we named pYH156_sgRNA_lib. The quality of the sgRNA plasmid library was verified by high-throughput sequencing. All 188 sgRNAs were observed in the library.

In the second step, all promoter sequences were amplified from the pooled plasmids of the *E. coli* promoter collection[21] using primers prYH070 and prYH071, and the vector backbone containing the sgRNA library was amplified from pYH156_sgRNA_lib using primers prYH072 and prYH073. These two PCR products were gel-purified and then ligated by Golden Gate assembly. The ligation products were purified using a DNA Clean & Concentrator kit (Zymo Research), and ~3.6 μg of purified ligation products were electroporated into 200 μL of fresh NEB 10-β competent cells using four electroporation cuvettes. The cells were plated on large LB agar plates (245 × 245 mm), resulting in about $8.2 \times 10^7$ individual clones. Transformants were scraped from the large agar plates, and the resulting combinatorial library plasmids (pYH160, Supplementary Data 10) were extracted using a maxiprep kit. Purified pYH160 plasmids (1 μg) were further electroporated into *E. coli* strain FR-E01, yielding >$10^8$ transformants. Transformed cells were resuspended in LB medium and then were used to prepare 2 mL glycerol stocks.

## Sorting the combinatorial library

Sorting experiments were performed in triplicate using cultures prepared in different weeks. For each replicate, 100 μL of the combinatorial library glycerol stock was thawed and inoculated in 250 mL of LB medium. When an $OD_{600}$ of 0.5 was reached, 1 mL of cells was diluted into 25 mL of a target medium (either M9 glucose, M9 glycerol, or LB medium). After a few hours of adaptation, 500 μL of the cultures were added to 50 mL of the target medium containing 1 μM aTc as inducer. The induced cells were grown at 37 °C until $OD_{600}$ reached 0.1–0.2. At this point, the cell cultures were supplemented with 250 μg/mL chloramphenicol to halt protein production and were kept on ice until sorting.

Cell sorting was performed on either FACSAria II (for cells grown in the M9 glucose) or FACSMelody (for cells grown in LB and M9 glycerol media) cell sorters (BD Biosciences). To control extrinsic protein production noise, events were gated around the mean fluorescence of mCherry, which is constitutively expressed on the reporter plasmid. Cells were sorted into 16 equally sized contiguous bins according to their GFP fluorescence intensity on a log scale[23,76] using a four-way purity sorting mode. The number of bins was chosen by considering both sorting time (5–10 h) and expression level difference between adjacent bins (less than 1.7-fold change). Both previous sort-seq experiments[17,22,76,77] and simulations[23] have shown that the use of 16 bins in our case allows reliable quantification of mean gene expression level. For each replicate, the flow rate during cell sorting was kept constant, and each bin was sorted for the same amount of time so that the number of cells collected in each bin was proportional to the phenotypic density in the population[77]. The numbers of cells sorted in each bin for each replicate were recorded for normalization in data analysis.

## Sample preparation for NGS

In each bin, sorted cells were added to an equal volume of LB medium with 50 ng/μL kanamycin and were grown overnight. Plasmids were extracted from 3 mL of overnight cell cultures using miniprep kits. From each bin, 50 ng of purified plasmids were amplified using the KAPA HiFi PCR Kit (Kapa Biosystems) for 20 cycles, using primers prYH071 and prYH087. The PCR products were purified using DNA Clean & Concentrator kits (Zymo Research) and then ligated to Illumina sequencing adapters via Golden Gate assembly. The adapter-labeled products were gel-purified to select DNA sizes between 400 bp and 1.5 kb. To enrich ligated products, gel-purified DNA products were subjected to another round of PCR using primers prYH128 and prYH129 for 8 cycles. Amplified adapter-labeled samples from each bin were then mixed in ratios that ensured that the number of reads was proportional to the number of cells sorted in each bin. The pooled sample was sequenced using partial lanes on a NovoSeq S4 XP Platform (2 × 150) at the Genome Technology Access Center of the McDonnell Genome Institute.

## NGS data processing

Paired-end reads were separately aligned to the pre-defined sgRNA library and the complete genome of *E. coli* MG1655 (U00096.3) using Bowtie2 v2.3.5[78]. For each promoter-end read, the genomic coordinates from the alignment were used to find the closest downstream operon and the closest downstream gene in the genome using BED-Tools v2.29.2[79]. Some non-promoter sequences whose end cannot be mapped to the first 200 bp of the coding sequence of the first gene in the operon were excluded for subsequent analysis. The remaining promoters are listed in Supplementary Data 2.

For each variant $i$, its read counts $r_{ij}$ in each bin $j$ were multiplied by $C_j/R_j$ to estimate its cell counts $c_{ij}$ sorted to bin $j$, where $C_j$ and $R_j$ are the number of cells collected in bin $j$ and the number of reads sequenced with barcode associated with bin $j$ respectively. This normalization step allows comparisons between bins after post-sort

growth, plasmid extraction, and NGS preparation by assuming the ratio of each variant in a bin does not change significantly. The fraction of cells of variant $i$ being sorted into bin $j$ is $f_{ij} = c_{ij}/\sum_j^{16} c_{ij}$. Due to the technical noise in sorting, a noise reduction method was applied by calculating adjusted cell fraction $f_{ij}^{adj} = (f_{ij} - \varepsilon)/\sum_j^{16}(f_{ij} - \varepsilon)$, where $\varepsilon = 0.05$ is a hyperparameter representing the noise background. To avoid negative values, the probability of a cell of variant $i$ truly coming from bin $j$ before sorting was estimated as $p_{ij} = \max(0, f_{ij}^{adj})/\sum_j^{16} \max(0, f_{ij}^{adj})$. It was assumed that the fluorescence distribution for each variant approximates a log-normal distribution[80–82] (Supplementary Figs. 1a and 2).

Parameter estimation was performed following previously described methods with minor modifications[22,23]. To find parameters $\mu_i$ and $\sigma_i$ for the log-normal distribution of each variant $i$, we used the Nelder-Mead method (Scipy package) to maximize the log-likelihood function:

$$\log L\left(\mu_i, \sigma_i, | , p_{ij}\right) = \sum_{j=1}^{16} p_{ij} \log(F_{\mu_i, \sigma_i}\left(B_j\right) - F_{\mu_i, \sigma_i}(B_{j-1})), \qquad (1)$$

where $F_{\mu_i, \sigma_i}$ is the cumulative distribution function of a normal distribution with mean $\mu_i$ and standard deviation $\sigma_i$, and $B_j$ and $B_{j-1}$ are the upper and lower boundaries at log scale of the bin $j$. Since we did not set the lower boundary for the first bin and the upper boundary for the last bin, $F_{\mu_i, \sigma_i}(B_0) = 0$ and $F_{\mu_i, \sigma_i}(B_{16}) = 1$. The mean GFP intensity of variant $i$ is then $\overline{GFP}_i = \exp(\mu_i + \sigma_i^2/2)$.

Kullback-Leibler (KL) divergence between the inferred distribution and $p_{ij}$ was calculated to evaluate the goodness of fitting for each variant $i$:

$$K\left(p_{ij}, \mu_i, \sigma_i\right) = \sum_{j=1}^{16} p_{ij} \log\left(p_{ij}\right) - \log L\left(\hat{\mu}_i, \hat{\sigma}_i, | , p_{ij}\right). \qquad (2)$$

To control the fitting quality, fitted parameters of variants with any of the following features were set as "not available (NA)" for subsequent analysis: (1) variants with mean GFP intensity not within our detection limits ($10^{1.5}$ to $10^5$ for the M9 glucose growth condition and $10^{1.5}$ to $10^{5.5}$ for the LB and M9 glycerol growth conditions); (2) variants with estimated cell counts $\sum_j c_{ij}$ less than 1 (Supplementary Fig. 14a); (3) variants with the KL divergence greater than 1 (Supplementary Fig. 14b); (4) variants with all cells in a single gate. These filters improved consistency among replicates (Supplementary Fig. 14c).

Data in each replicate was processed using the above procedures independently. To reduce the systematic differences between replicates, we applied linear transformation to mean GFP intensity measured from replicate #1 and replicate #2 using scale of replicate #3. The mean $M_i$ and standard deviation $S_i$ of rescaled $\log(\overline{GFP}_i)$ of variant $i$ between replicates were calculated. For negative control variants, their mean $M_{i0}$ and standard deviation $S_{i0}$ of rescaled $\log(\overline{GFP}_{i0})$ were obtained by treating negative control sgRNAs (NC_35, NC_82, NC_84, and NC_89) in replicates as independent samples. Data from sgRNA NC_31 is inconsistent with data from the other four negative control sgRNAs, therefore, NC_31 was excluded from data analysis. For each promoter, outliers in negative control samples were excluded using the interquartile range (IQR) method. Variants with $S_i$ greater than 0.7 were also excluded to ensure the data consistency.

### Differential expression analysis
We adopted a method of mean comparison for log-normal distribution[83] to determine whether the perturbed activity of a promoter by TFKD for variant $i$ was significantly different from the activity of the promoter measured by negative control samples $i0$. Z tests were performed using the Z score calculated by

$$Z_i = \frac{M_i - M_{i0} + (1/2)\left(S_i^2 - S_{i0}^2\right)}{\sqrt{\frac{S_i^2}{n_i} + \frac{S_{i0}^2}{n_{i0}} + (1/2)\left(\frac{S_i^4}{n_i-1} + \frac{S_{i0}^4}{n_{i0}-1}\right)}}, \qquad (3)$$

where $n_i$ and $n_{i0}$ are the number of qualified samples of TFKD variant $i$ and its corresponding negative control $i0$ with the same promoter. To control the false discovery rate (FDR), $q$ values were calculated based on the $P$-value from the Z tests[84]. Given that promoter activity in the negative control is not consistent with the median promoter activity for a small number of promoters, fold changes relative to both negative control activity and median promoter activity need to be larger than 1.7 to call them substantial effects. Analyzed differential expressions can be found in Gene Expression Omnibus (GEO) with access number GSE213624. Functional annotation and enrichment analysis were performed using the DAVID web server[85].

### Reverse transcription-qPCR
Triplicate colonies were grown overnight in LB. Cultures were then diluted by 200-fold into 5 mL M9 glucose medium and grown for 1 h. Cells were then induced with 1 μM aTc and grown for an additional 2 h. Cultures were then diluted 900-fold in M9 glucose containing 1 μM aTc and grown to OD$_{600}$ of 0.1, followed by RNA extraction using 2 mL culture (Zymo Research Quick-RNA kit). All the RNA samples were then treated with DNAse (Zymo Research) and reverse transcribed to cDNA with RevertAid First Strand cDNA Synthesis Kit (Thermo). The cDNA samples were then subjected to qPCR using the PowerTrack SYBR Green Master Mix (Thermo) and a Quantstudio 3 instrument (Applied Biosystems). The constitutive gene *dnaK* was used as an internal control, and fold change for each gene was calculated using the $2^{-\Delta\Delta CT}$ method[86].

### Kinetic assays for a subset of individual variants
We individually constructed a subset of the combinatorial library consisting of five promoters (P$_{fadE}$, P$_{glyA}$, P$_{lacZ}$, P$_{marR}$, and P$_{metA}$) and ten sgRNAs targeting nine TF genes (*acrR*, *arcA*, *crp*, *fadR*, *lacI*, *marA*, *marR*, *metJ*, and *purR*) and a negative control (NC_84). These plasmids (Supplementary Data 10) were confirmed by Sanger Sequencing and were individually transformed into *E. coli* strain FR-E01. Single colonies were inoculated into 0.5 mL of LB medium with 25 ng/μL kanamycin in a 96-well deep-well plate and grown overnight. The overnight cultures were diluted 1:255 into 150 μL of M9 glucose medium in a 96-well plate. After 3 h, the cultures were diluted 1:900 into 150 μL of M9 glucose medium with 1 μM aTc, and then incubated in an Infinite 200 Pro plate reader (Tecan) at 220 r.p.m. and 37 °C. OD$_{600}$ and GFP fluorescence were measured every 10 min over 10 h. Customized MATLAB scripts were used for data processing, including the background correction and OD$_{600}$ normalization. GFP/OD$_{600}$ values for each strain remained nearly constant when OD$_{600}$ (converted to the equivalent value for 1-cm pathlength measurements) was between 0.08 and 0.32. The steady-state GFP expression levels were calculated by averaging GFP/OD$_{600}$ values from the two closest measurements above and below OD$_{600}$ = 0.2 for all strains.

### Promoter activity measurements in TF-tunable strains
Reporter plasmids containing selected promoters for validation were obtained from the promoter library[21], then transformed into TF-tunable strains from the Brewster lab[34] (Supplementary Data 11). We noticed that the expression level of MetR-mCherry was as low as our detection limit and could not be induced by aTc in the MetR-tunable strain. Due to this loss of tunability, reporter plasmids in the MetR-tunable strain were also transformed into a control strain with the wild-type MetR expression level for comparison.

Promoter activities were measured using the method described in the "Kinetic assays for individual variants" section with the following modifications. First, to investigate condition-specific perturbation effects, some strains were grown in M9 media with different carbon sources or metal ions. The PdhR-tunable strain harboring the reporter plasmid for the *fadE* promoter was grown in M9 minimal media supplemented with one of the following carbon sources: 0.4% glucose, 0.2% succinate, 4 mM oleate, or 0.5% glycerol plus 4 mM oleate. Strains harboring the *arnB* promoter reporter plasmid were tested in M9 glucose media with either 0.4 mM $FeSO_4$ or 0.2 mM $Fe_2(SO_4)_3$. Second, to generate different TF expression levels, the aTc inducer was added in concentrations of 0, 2, and 20 nM. Third, for cell cultures that contained $FeSO_4$ or $Fe_2(SO_4)_3$, steady-state periods were identified by examining the $GFP/OD_{600}$ and $OD_{600}$ data because $OD_{600}$ measurements were affected by $FeSO_4$ or $Fe_2(SO_4)_3$, especially when $OD_{600}$ was low.

### Analysis of TF binding sites from DAP-seq

All the BED files for TF binding peaks in *E. coli* identified from DAP-seq[42] were screened, merged, and mapped to *E. coli* promoters investigated in this study using BEDTools v2.29.2[79]. All binding sites associated with TF-promoter pairs missing in our PPTP-seq dataset were excluded from subsequent analysis. TSS information was obtained from RegulonDB v 10.0[1]. TF concentration was estimated from the ribosome-profiling results[87].

The center position of each TF binding site relative to the TSS was calculated as the relative position to TSS. For promoter regions with multiple TSSs, only the TSS closest to the start codon of the downstream gene was analyzed. The binding strength between a TF and its binding site is defined by fold enrichment over the background in DAP-seq experiments. If a promoter had multiple binding sites for a TF, only the binding site with the highest binding affinity was analyzed. The relative binding strength per TF was calculated as the fold enrichment for the TF over the background, divided by the maximum fold enrichment for the TF. The relative local binding strength was calculated as the fold enrichment for a TF bound on a promoter over the background, divided by the maximum fold enrichment for all TFs bound on the promoter.

### Statistics and reproducibility

No statistical methods were used to predetermine the sample size. PPTP-seq experiments were performed in three biological replicates for M9 glucose condition and two biological replicates for M9 glycerol and LB conditions to assess the reproducibility of these measurements. The means and standard deviations between replicates were calculated and used in statistical analysis. Sequencing reads for cells sorted into bin #1 after growth in M9 glycerol medium were excluded from data analysis due to potentially unwanted mutations. Data exclusion after log-normal distribution fitting is described in "Methods: NGS data processing".

### Reporting summary

Further information on research design is available in the Nature Portfolio Reporting Summary linked to this article.

## Data availability

The PPTP-seq generated in this study has been deposited in the GEO database under accession code GSE213624. The plate reader and RT-qPCR data generated in this study are provided in the Source Data file. The processed data of RNA-seq[30] and EcoMAC microarray[32] used in this study are available at GitHub [https://github.com/CovertLab/wcEcoli/tree/master/reconstruction/ecoli/flat]. The DAP-seq data are available in the Supplementary Data 1 file in ref. 42 [https://doi.org/10.1038/s41592-021-01312-2]. The other TF binding datasets used in this study are available at RegulonDB high-throughput collection [https://regulondb-datasets.ccg.unam.mx/ht/tfbinding/#/]. Source data are provided with this paper.

## Code availability

Scripts and Jupyter Notebooks are available at https://doi.org/10.5281/zenodo.8309683.

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

## Acknowledgements

The authors would like to thank M. Brent, B. Cohen, A. Schmitz, C. Hartline, and G. Urtecho for thoughtful discussion; C. Zou for construction of a part of the pYH156 region; E. Lantelme, H. Feng, K. Kim, N. Urs AN, and H. Zaher for technical assistance with FACS; E. Martin for managing high-throughput computing facility; V. Parisutham and R. Brewster for providing the TF-tunable *E. coli* strains; and J. Ballard for editing the manuscript. This work was supported by the National Institute of General Medical Sciences of the National Institutes of Health (R35GM133797). Y.H. is supported by a T32 HG000045 training grant from the National Human Genome Research Institute.

## Author contributions

Y.H. and F.Z. conceived the project, designed the experiments, analyzed the data, and wrote the manuscript. Y.H. performed all experiments and processed the data. W.L. and J.L. helped with cloning and plate reader experiments. W.L. helped with testing PPTP-seq in LB and M9 glycerol media. A.F. performed RT-qPCR experiments.

## Competing interests

The authors declare no competing interests.
