## [Peer Review File · Nature Communications]

Reviewers' Comments:

Reviewer #1:

Remarks to the Author:

Review of "Genome-wide promoter responses to CRISPR perturbations of regulators reveal regulatory networks in Escherichia coli" by Y. Han, W. Li, J. Li and F. Zhang.

General comments: This manuscript attacks an important problem in modern genomic science: what is the nature of the regulatory networks that preside over the 1000s of genes within a given organism. Surprisingly, even in *E. coli*, we remain vastly ignorant of how its genes are regulated. The manuscript by Han et al. presents a new method for studying transcriptional regulation called PPTP-seq (pooled promoter responses to TF perturbation), which is used to study promoter activity in *E. coli*. The method captures a large number of TF-promoter interactions in a single experiment (all *E. coli* TFs, and a large proportion of *E. coli* promoters). However, in our opinion, the paper falls far short of what it could have done because they focused on only a single growth condition. Further, we found their attitude about the general topic of how environmental signals couple to gene expression to be rather cavalier.

We are happy to see this paper published, and offer our comments really in the spirit of take-it-or-leave-it. We are definitely people that are interested in seeing work like this succeed, but have the sense that the authors are not thinking about how to make their results useful for databases such as RegulonDB and Ecocyc.

Overarching Comments:

- Very detailed explanation of methods and procedure.
- Plate reader validation helps increase confidence in their pooled measurements.
- Only one growth condition is used (Minimal media with glucose), while it is claimed to be a simple assay (taking less than two weeks), this raises the question of why they didn't take a few more weeks and dig into some of the growth conditions used by people like Schmidt/Heinemann or the recent Balakrishnan/Hwa Science paper. This would have made this work much more compelling. In our opinion, the conclusions drawn from this dataset can only be made in regard to growth in minimal media with glucose. Yet, Schmidt/Heineman make it very clear that depending upon growth conditions, different genes are differentially expressed and as is clear from the carbon transporters, if one does not perturb with something a given gene cares about, there will be no effect. These considerations are not even taken into consideration in the discussion. Any quantitative conclusions, such as the number of promoters regulated by a single TF or the number of promoters that do not have a TF associated with, rely on the growth conditions tested. The most prevalent example would be the global activator CRP, which only binds to many of its binding sites (e.g. in the lac operon) in the absence of glucose. Hence, these binding sites could not have been found in this experiment, unless the growth condition is varied. (it is mentioned briefly in line 154, but needs to be made more specific)
- Why were no DNA barcodes used? One could barcode both libraries and assemble them with restriction digest cloning, which would allow one to only sequence barcodes. This question is more for self interest and not a critique.
- The paragraph about the discrepancy between the results from this paper and DAP-seq are very interesting and deserve a lot more attention. How is the DAP-seq experiment performed and what could cause these discrepancies. We interpret figure 5 to show us that none of the suggested possibilities clearly explain the differences.
- This method relies on TFKD by CRISPRi, but this is only measured indirectly via changes in each promoter's reporter activity. A direct measurement (such as by RNA-seq) of TF expression levels could be a useful supplementary experiment. For example, not many TFKDs were found to have significant effects (Fig 5b), but could this be due to e.g. poor or variable KD efficiency, which is not further examined.
- Some of the figures in the paper have so many datapoints that are poorly organized, which makes it difficult for readers to extract key information. The authors could consider visual representations that more directly convey the key data points and remove information that are irrelevant to the main points that the authors are trying to make. For example, in figure 3a, it is unclear whether the readers are supposed to recognize patterns in the network or simply come to a quantitative interpretation about the number of genes are that under different types of

regulation. In the former case, the authors should consider ordering the genes in such a way that makes the pattern more apparent; in the latter case, a bar chart or a pie chart might convey the message more effectively. Another figure requires the reader to do unnecessary mental maneuver is figure 4a. Since the paper does not seek to educate the readers on the exact regulatory network of one-carbon metabolism, it might be more effective to only show key regulatory interactions in the network instead of asking readers to seek them out from such a comprehensive map.

Specific comments/nitpicks:

- Line 37: Need reference for number of confirmed TFs and operon controlling promoters
- Line 38: "500,000 possible" – explain this please.
- Line 40: "mapping genome-scale TRN for all TFs.." what is meant by mapping? Personally, I find this to be nonsensical. Mapping to us means that RegulonDB could have a full entry in which we know precisely where the binding sites are, which transcription factors bind them and what effectors interact with those transcription factors. Though the language of this paper is grandiose, the results are quite modest in comparison with that very hard challenge.
- Line 42: "allow multiplexed mapping" – again, we seem to have very different ideas about what it means to map networks. Uri Alon's book shows what it means to have a map, or Eric Davidson's biotapestry.
- Line 46: we completely object to the comment "While powerful, this method alone..." First, this shows a lack of scholarship. Several of those papers go all the way from complete regulatory ignorance to having a full description of the promoter architecture, what transcription factors bind the binding sites, what the energies of interaction are between the transcription factors and the DNA are. Hence, those results: a) can be input into RegulonDB and b) are used as the basis of design of new circuits. Our sense is that these authors are attempting (and spuriously) to set up some straw man to make their own work seem better, and frankly, this just erodes their credibility because either they didn't understand this other work, or they are distorting it to make their own work seem more important. Pretty annoying.
- Line 53: "lasting just two weeks" – what does this even mean? And, if it takes just two weeks, then why not taking advantage of the important work done by Schmidt, by Hwa, by others who have demonstrated very clearly that changing the growth conditions changes the level of expression.
- Line 76: 16 bins – no justification. The work from Justin Kinney (PNAS) and then Belliveau examined the question of binning. AT the very least, some attempt at scholarship and caution could be helpful.
- Line 78: "NovoSeq" should be spelled "NovaSeq"
- Lines 97-105: There are five combinations that could not be recovered. Add a clarifying statement about these 5 (why were they not able to be recovered)
- Line 128-129: This sentence has redundant information with the previous sentence.
- Lines 134-139: so facile and really, we find, sloppy. Especially the part about mutual regulation and repressilators. Again, the same complaint. You can't use one condition and expect to find what kinds of regulatory interactions are present. A reasonable guess is that more than 50% of transcription factors are allosteric (or inducible) and hence, if you are not using the thing that induces a given TF, it will not reveal itself.
- Line 143-146: In which growth conditions were these autoregulated TFs identified. If different from minimal media with glucose, this comparison cannot be made
- Line 152-153: These authors can say what they want, but for readers like us, such sloppy remarks erode their credibility. How based on one set of conditions are you prepared to make such big remarks about TF autoregulation? It just seem that these authors want to set up straw men and then knock them down without a careful and thorough analysis. Again, say what you want. But know that the four of us that read and reviewed the paper feel less convinced and less supportive rather than more when such grand statements are made based on so little evidence.
- Line 193: We have know idea what "relatively stronger local binding strength" – relative to what? What is this about?
- Line 209-211: "48% of promoters were regulated by more than one TF", but earlier you said that 31% (line 117) were regulated by one activator or one repressor. On top of that, 62% of all promoters were variable. These numbers do not add up, please clarify.
- Line 215-217: Please refer to Rydenfelt, Mattias, et al. "The influence of promoter architectures and regulatory motifs on gene expression in Escherichia coli." PLoS one 9.12 (2014): e114347.
- Line 218-228: please consider referring to Garcia & Phillips, 2011; Razo-Mejia et al., 2018, Cell Systems, Belliveau et al., 2018, PNAS; Barnes et al., 2019, PLOS Comp Bio

- Line 239-241: Refer to other growth conditions. Although work in this study does uncover information about individual TFs, these findings do not qualify as “unknown regulatory rules”.
- Line 250-251: This is no different to other methods such as deep mutational scanning
- Line 273-276: What is strong evidence? And why were 183 picked, when 181 had “strong evidence”?
- Discussion: Discuss possible complementary uses with other methods
- Figure 1d: It is impossible to see the gray dots outside of the 2-fold change line, which we think are supposed to be the most significant ones?
- Figure 2a: Isn't this the same data as in Figure 1d, but replotted? Figure 1d,
- Figure 2 b-c: Barplot on a log scale is very hard to interpret. The x-axis labels are backwards and should be “the number of promoters *regulating* a specific gene” and “the number of TF genes *regulated by* a specific promoter.
- Generally, the figures are just a big disappointment. For example, how do these authors expect people like us that REALLY care about the nature of the regulatory linkages in E. coli and other bacteria, to truly learn something from Fig. 3a? It feels as though no attempt was made to be creative or pedagogical or helpful to people like Collado that do RegulonDB to make figures that help us understand in actionable terms what is learned about each gene. Sure, it is a lot of work, but for example, maybe they need in the SI or a website to take EVERY gene they looked at and make a cartoon that SIMPLY summarizes what they think happened with that gene. For example, earlier in the paper, they made a claim about simple repression and simple activation being the most common motifs. Great, so why not make a few tables that say these are ALL the genes we found are regulated by simple activation and here is the name of the TF that does the activation? Here are ALL the genes we found that are regulated by simple repression and here is the name of the TF that does the repression?

Though we fear the editors will take our remarks as the basis for rejection, that is NOT what we are arguing. This is a very interesting approach, and seems to have led to some insights. But, for slow thinkers like us, why can't these authors work harder to not fall into the typical trap of modern genomic science with figures that are largely useless and instead really trying to mine their gold mine of data in ways that make it accessible in a practical way?

Reviewer #2:

Remarks to the Author:

In the present manuscript, Han Y. et al developed a new technique called pooled promoter responses to TF perturbation sequencing (PPTP-seq) to measure the activity of 1373 E. coli promoters under single knockdown of 183 TFs via CRISPR interference and illustrated more than 200,000 possible TF-gene responses in one experiment. The PPTP-seq enabled the authors to identify novel TF auto-regulatory responses and also novel transcriptional control in one-carbon metabolism. Finally, by comparing their data to the published TF binding peaks identified by in vitro DAP-seq (PMID: 34824476), the authors found that relative binding strengths of TFs on a promoter were a major determinant of promoter response. In general, this study showed the power of the PPTP-seq in the study of high throughput TF-gene responses in E.coli. However, several concerns need to be clarified.

1. It is important to know the repression efficiency of CRISPR interference on the expression of selected TFs. However, this paper lacks such knowledge.
2. In line 75-76, transformed cells were grown in minimal glucose medium to a steady state and sorted into 16 bins based on their fluorescence intensity (Supplementary Fig. 1a). The authors should give reasons that why sort the cells into 16 bins (why not more bins) and describe how they quantify the expression in the main text in more details. It is difficult to understand how they did the quantification as described in the method. Also it is necessary to give the fold changes (non-targeting sgRNA vs a sgRNA targeting) in Supp. Fig 2. In Fig. 2a, most of the Log2 fold changes of the varied TF-gene pairs were smaller than 4. Especially, the fold change of the varied LacI-LacZ pairs was much smaller than the reported number. The resolution of the expression quantification of the promoter could be quite low and the rather low resolution of the expression quantification may hide more varied TF-gene pairs.
3. In line 85-87, out of 182 tested TFs, 178 of them were able to activate or repress at least one

promoter. The author should compare their data to the published data set to validate their findings. In Fig 2d, it is also necessary to compare the variable and constant promoters identified here to the published data.

4. In Fig 3a, the regulatory network looks very complicated, and this panel was even not cited in the main text. The authors may find a way to highlight the most important one and also compare this finding to the published data.

5. In line 176-177, these previously unidentified MetR regulations were further verified using the tunable TF library (Fig. 4c). However, this was done in the MetR knockdown background. Are the newly identified MetR targets also identified by the reported *in vitro* DAP-seq?

6. How much can we trust the results reported in Fig 5c-f, since only the regulatory or non-regulatory information were obtained in this work *in vivo*, while the other important parameters including TF concentration and binding strength were obtained by other labs *in vitro* or in different conditions?

Reviewer #3:

Remarks to the Author:

In their manuscript "Genome-wide promoter responses to CRISPR perturbations of regulators reveal regulatory networks in *Escherichia coli*" Han et al. describe PPTP-seq, a pooled high-throughput method of evaluating promoter activity in the presence of CRISPRi knockdowns of transcription factors. The authors work represents a major advance. However, additional work needs to be done to characterize the performance of PPTP-seq and to better connect it to previous methods. I outline these issues in the text below:

Major points.

1. PPTP-seq fundamentally provides the same kind of information as ChIP-seq, DAP-seq, SELEX, RNA-seq, and many other methods. Therefore, it is important to provide a more thorough comparison of its pluses and minuses of their method compared to the others. Though the authors provide a comparison with DAP-seq, this is not enough, especially given how well characterized many *E. coli* TFs are.

a. After reading the manuscript, I have no idea what the expected false positive and false negative rates are, nor the distribution between direct and indirect regulatory interactions among the results.

b. How do your steady state promoter activity measurements compare to RNA-seq and/or nascent RNA-seq measurements of the same?

2. Sequence analysis of promoters would be useful for understanding the nature of observed regulatory interactions. How often is it the case that a TF knockdown affects a promoter, but no binding site for that TF is apparent at that promoter?

3. Several aspects of the paper are insufficiently described. For example:

a. A strong point of the paper is that the authors used the Alon collection of promoters. However, some basic summary about how these promoters were chosen and how much of the sequence they encompass is really important for interpreting the authors' results and shouldn't require readers to find the Alon paper. Listing all of the promoters in a supplementary table would also be helpful.

b. What CRISPRi system did the authors use? How was it induced, and what was the approximate knockdown of the TFs?

c. Since DAP-seq plays an important role in the paper, perhaps a few words about what it is may help readers understand what's going on.

4. The authors should address the issue of CRISPRi polarity, especially since bacterial TFs are

frequently located near the genes they regulate. For example, marR is located immediately upstream of its targets marA and marB. Similarly, nagC is located in the middle of its operon, and mraZ is located at the start of the (highly essential) dcw operon.

5. Most sequencing based assays compare the relative abundance of different strains . In contrast, this analysis relies on accurately quantifying absolute abundance. Quantifying absolute abundance allow direct comparisons between bins. However, in order for the method to be valid, the authors need to demonstrate that the same number reads reflect the same number of cells in each bin. It is not clear to me how the authors assure this through the different plasmid extraction and library prep steps. A simple method for normalizing reads would be to sequence an unsorted (but otherwise identically treated) sample of the library and then apply a linear regression to determine relative weights for each of the bins. ($\text{unsorted_reads} = w_1 \cdot \text{bin}_1 + w_2 \cdot \text{bin}_2 \dots \text{etc}$).

Minor points

1. L19 – you are not measuring “phenotypic responses”, but gene expression.
2. L32 – “Information about bacterial cellular responses is mostly encoded in promoters...” This is an overly broad statement with no supporting information that serves only to antagonize people who study non-transcriptional regulation. Plenty of bacterial responses involve primarily non-transcriptional responses.
3. L46-L47 – “this method...cannot provide information about which TF is regulating the promoter.” Neither can PPTP-seq, since knockdown of a TF can cause compensatory responses that involve other TFs.
4. L57-59 – “Our study uncovered novel regulatory rules, including TF autoregulatory responses, complex transcriptional control of metabolic pathways, and promoter responses to coregulation.” You did not discover these “rules”, you discovered some (minor) examples that further reinforce these well-established principles.
5. L120-121 – “COG analysis”. Consider also doing the analysis using GO terms and KEGG categories. You may find more descriptive/useful functional annotations.
6. L143 – PgrR is known to autoregulate. They even did a gel shift experiment.
<https://doi.org/10.1111/gtc.12026>

Reviewer #1

Review of “Genome-wide promoter responses to CRISPR perturbations of regulators reveal regulatory networks in Escherichia coli” by Y. Han, W. Li, J. Li and F. Zhang.

General comments: This manuscript attacks an important problem in modern genomic science: what is the nature of the regulatory networks that preside over the 1000s of genes within a given organism. Surprisingly, even in *E. coli*, we remain vastly ignorant of how its genes are regulated. The manuscript by Han et al. presents a new method for studying transcriptional regulation called PPTP-seq (pooled promoter responses to TF perturbation), which is used to study promoter activity in *E. coli*. The method captures a large number of TF-promoter interactions in a single experiment (all *E. coli* TFs, and a large proportion of *E. coli* promoters). However, in our opinion, the paper falls far short of what it could have done because they focused on only a single growth condition. Further, we found their attitude about the general topic of how environmental signals couple to gene expression to be rather cavalier.

We are happy to see this paper published, and offer our comments really in the spirit of take-it-or-leave-it. We are definitely people that are interested in seeing work like this succeed, but have the sense that the authors are not thinking about how to make their results useful for databases such as RegulonDB and Ecocyc.

Overarching Comments:

- Very detailed explanation of methods and procedure.
- Plate reader validation helps increase confidence in their pooled measurements.

We thank the reviewer for the positive evaluation of our method and critiques on our manuscript. During revision, we performed PPTP-seq for two additional conditions, analyzed and compared the data collected from different conditions. We also revised the figures and updated the tables to make the results more easily trackable to readers and other databases.

- Only one growth condition is used (Minimal media with glucose), while it is claimed to be a simple assay (taking less than two weeks), this raises the question of why they didn't take a few more weeks and dig into some of the growth conditions used by people like Schmidt/Heinemann or the recent Balakrishnan/Hwa Science paper. This would have made this work much more compelling. In our opinion, the conclusions drawn from this dataset can only be made in regard to growth in minimal media with glucose. Yet, Schmidt/Heineman make it very clear that depending upon growth conditions, different genes are differentially expressed and as is clear from the carbon transporters, if one does not perturb with something a given gene cares about, there will be no effect. These considerations are not even taken into consideration in the discussion. Any quantitative conclusions, such as the number of promoters regulated by a single TF or the number of promoters that do not have a TF associated with, rely on the growth conditions tested. The most prevalent example would

be the global activator CRP, which only binds to many of its binding sites (e.g. in the lac operon) in the absence of glucose. Hence, these binding sites could not have been found in this experiment, unless the growth condition is varied. (it is mentioned briefly in line 154, but needs to be made more specific)

We thank the reviewer for the helpful comments. To make the work more compelling, we followed the reviewer's suggestion and performed PPTP-seq for two additional growth conditions: exponential growth in LB rich medium and exponential growth in minimal glycerol medium. Indeed, we observed a substantial number of regulations specific to each growth medium. Meanwhile, we also observe many regulations that are present in two or all three growth conditions. These new results are incorporated in Figures 6, Supplementary Fig 12, and Supplementary Tables 2, 3, and 6. Additionally, extensive discussions were added to comment on the environmental influence of gene expression at the systems level, lines 267-288.

- Why were no DNA barcodes used? One could barcode both libraries and assemble them with restriction digest cloning, which would allow one to only sequence barcodes. This question is more for self interest and not a critique.

Our PPTP-seq method is compatible with barcoded reporter assays. The reason we did not use barcode is to avoid complications in promoter library cloning. The promoter library we used was taken from an existing plasmid library¹. Adding barcodes to this library will involve additional cloning steps that may change promoter activities.

- The paragraph about the discrepancy between the results from this paper and DAP-seq are very interesting and deserve a lot more attention. How is the DAP-seq experiment performed and what could cause these discrepancies? We interpret figure 5 to show us that none of the suggested possibilities clearly explain the differences.

We thank the review for raising this point. We discussed how DAP-seq was performed and provided explanations to these discrepancies in the revised manuscript (lines 205-206, 227-233 and 237-238). Briefly, the discrepancies can be caused by indirect functional regulation, low TF expression levels, low TF activity (affected by other molecules), and/or complex regulatory patterns. Fig. 5 showed that some potentially bound but not regulating TFs have relatively lower expression level (Fig 5e, f) and binding strength (Fig 5i) compared to other TFs at the same promoter. Additionally, individual examination of *arnB* and *fadE* promoters revealed that TF activity and context-dependence (presence of other active TFs) matters (Supplementary Figure 10).

- This method relies on TFKD by CRISPRi, but this is only measured indirectly via changes in each promoter's reporter activity. A direct measurement (such as by RNA-seq) of TF expression levels could be a useful supplementary experiment. For example, not many TFKDs were found to have significant effects (Fig 5b), but could this be due to e.g. poor or variable KD efficiency, which is not further examined.

Thank you for the suggestion. We used two methods to evaluate the repression efficiency of CRISPRi in our library and the results were added to the revised manuscript. First, we examined sequences in

the promoter library that can be directly targeted by sgRNAs, thus the CRISPRi repression efficiency of a TF can be estimated by the fold of repression on its own promoter. Totally 41 TFs have their promoters in our library, and 39 of them (95%) showed repression (Student's t-test P value < 0.05) (Supplementary Fig 5a). Second, additional RT-qPCR experiments were performed to examine the CRISPRi repression efficiency. The nine TFs that were used to examine individual responses (Supplementary Fig 6) and another five TFs (HipB, NarP, OmpR, YefM, and YeoB) randomly selected were chosen as representatives. The RT-qPCR results showed clear repression for 12 out of 14 TFs, representing 86% is effective (Supplementary Fig 5b). We further found a clear negative correlation between the degree of CRISPRi repression and TF expression level measured from TF promoter activity (Supplementary Fig 5c, d). This explains the lack of repression for the small fraction of TFs (e.g., *qseB* and *ttdR*).

Supplementary Figure 5. Evaluation of CRISPRi repression efficiency.

a and **b** Repression of TF genes by CRISPRi measured by PPTP-seq (**a**) and RT-qPCR (**b**). Data points show biological replicates. **c** Relative TF expression level as measured by their promoter activities in PPTP-seq using the strain with non-targeting control sgRNA. **d** Relationship between CRISPRi repression activity and TF promoter activity without CRISPRi repression.

We updated these results in main text (lines 105-112) and Supplementary Note 1.

- Some of the figures in the paper have so many datapoints that are poorly organized, which makes it difficult for readers to extract key information. The authors could consider visual representations

that more directly convey the key data points and remove information that are irrelevant to the main points that the authors are trying to make. For example, in figure 3a, it is unclear whether the readers are supposed to recognize patterns in the network or simply come to a quantitative interpretation about the number of genes that are under different types of regulation. In the former case, the authors should consider ordering the genes in such a way that makes the pattern more apparent; in the latter case, a bar chart or a pie chart might convey the message more effectively. Another figure that requires the reader to do unnecessary mental maneuver is figure 4a. Since the paper does not seek to educate the readers on the exact regulatory network of one-carbon metabolism, it might be more effective to only show key regulatory interactions in the network instead of asking readers to seek them out from such a comprehensive map.

Thank you for this suggestion. We revised both Figs. 3a and 4a to highlight the main points that we want to deliver.

In the revised Fig. 3a, we reordered the gene in the network based on whether they regulate other TFs or are regulated by other TFs. We also differentiate direct interactions from indirect interactions using solid or dashed lines according to the other reviewers' suggestions.

In the revised Fig. 4a, pathways and genes not mentioned in the manuscript were removed. We highlighted the key regulations revealed from this study in Fig 4a and added a more comprehensive map in the Supplementary Material for readers who would like to obtain a complete view of regulation in C1 metabolism.

Specific comments/nitpicks:

- Line 37: Need reference for number of confirmed TFs and operon controlling promoters

In RegulonDB 10.0, there are 183 TF-encoding genes and 2619 operons. We clarified this in the revised manuscript (lines 39-40).

- Line 38: “500,000 possible” – explain this please.

The ~500,000 possible TF-operon responses were calculated from 183 TFs and 2619 operons (183×2619). We explained this in the revised manuscript (line 40).

- Line 40: “mapping genome-scale TRN for all TFs..” what is meant by mapping? Personally, I find this to be nonsensical. Mapping to us means that RegulonDB could have a full entry in which we know precisely where the binding sites are, which transcription factors bind them and what effectors interact with those transcription factors. Though the language of this paper is grandiose, the results are quite modest in comparison with that very hard challenge.

Thank you for pointing this out. We changed the confusing wording. In the revised manuscript, we used the word “identify responsive target genes for all TFs...” to better reflect our work (line 42).

- Line 42: “allow multiplexed mapping” – again, we seem to have very different ideas about what it means to map networks. Uri Alon's book shows what it means to have a map, or Eric Davidson's biotapestry.

Thank you for pointing this out. We changed the wording to “systematic analysis of transcriptional response to various genetic perturbations” to describe the cited work more clearly (lines 44-45).

- Line 46: we completely object to the comment “While powerful, this method alone...” First, this shows a lack of scholarship. Several of those papers go all the way from complete regulatory ignorance to having a full description of the promoter architecture, what transcription factors bind the binding sites, what the energies of interaction are between the transcription factors and the DNA are. Hence, those results: a) can be input into RegulonDB and b) are used as the basis of design of new circuits. Our sense is that these authors are attempting (and spuriously) to set up some straw man to make their own work seem better, and frankly, this just erodes their credibility because either they didn’t understand this other work, or they are distorting it to make their own work seem more important. Pretty annoying.

To clarify “this method alone”, we changed to “this method alone without prior knowledge of TF binding sites” in the revised manuscript. What we mean is that results from promoter mutational scanning have to be used together with previous knowledge on TF binding (e.g., Belliveau et. al. and Ireland et. al. utilized additional DNA affinity chromatography and mass spectrometry to identify potentially bound regulatory proteins^{2,3}) to identify cis-regulatory elements. Without additional binding information, promoter mutational scanning alone cannot identify which TF is regulating. This has been clearly shown in Fig. 5D of a large-scale promoter mutational scanning study⁴.

- Line 53: “lasting just two weeks” – what does this even mean? And, if it takes just two weeks, then why not taking advantage of the important work done by Schmidt, by Hwa, by others who have demonstrated very clearly that changing the growth conditions changes the level of expression.

The “two weeks” timeline was estimated by the sum of each step involved in a PPTP-seq experiment: cell culture preparation takes two days; cell sorting takes two days; NGS library preparation from sorted cells takes two to three days; and sequencing takes one to two days. Of course, this timeline is estimated assuming no waiting time between any step.

We appreciate the reviewer’s suggestion and performed PPTP-seq for two additional growth conditions: exponential growth in LB medium and exponential growth in minimal glycerol medium. The results were added to Figures 6 and Supplementary Fig 12.

- Line 76: 16 bins – no justification. The work from Justin Kinney (PNAS) and then Belliveau examined the question of binning. At the very least, some attempt at scholarship and caution could be helpful.

The number of 16 bins was chosen from consideration of both sorting time and gene expression resolution, as well as our previous experience⁵, where we studied the effects of bin numbers to gene expression quantification. Previous sort-seq experiments from both ours and others⁵⁻⁹ have shown that mean gene expression can be reliably measured using 16 bins. More bins with the same number of sorted cells will result in longer sorting time. We explained the choice of 16 bins in the method of the revised manuscript (lines 415-418).

- Line 78: “NovoSeq” should be spelled “NovaSeq”

We corrected it.

- Lines 97-105: There are five combinations that could not be recovered. Add a clarifying statement about these 5 (why were they not able to be recovered)

Those five combinations are missing from our dataset due to their low read counts. We clarified this in the revised manuscript.

“The other five combinations were missing in all three replicates due to their low read counts.”

- Line 128-129: This sentence has redundant information with the previous sentence.

Thank you for pointing this out. We modified the sentence to remove the redundant information.

- Lines 134-139: so facile and really, we find, sloppy. Especially the part about mutual regulation and repressilators. Again, the same complaint. You can’t use one condition and expect to find what kinds of regulatory interactions are present. A reasonable guess is that more than 50% of transcription factors are allosteric (or inducible) and hence, if you are not using the thing that induces a given TF, it will not reveal itself.

Following the reviewer’s suggestion, we performed PPTP-seq for two additional conditions. We observed one mutual regulation and one repressilator in LB media, but no binding evidence was found for these regulations and suggesting indirect regulation. Further, these regulations were not found in minimal glycerol. We revised our main text (lines 162-164) and discussed these new results in Supplementary Note 3.

- Line 143-146: In which growth conditions were these autoregulated TFs identified. If different from minimal media with glucose, this comparison cannot be made.

We changed this sentence to clarify that these autoregulated TFs were identified in M9 glucose.

- Line 152-153: These authors can say what they want, but for readers like us, such sloppy remarks erode their credibility. How based on one set of conditions are you prepared to make such big remarks about TF autoregulation? It just seems that these authors want to set up straw men and then knock them down without a careful and thorough analysis. Again, say what you want. But know that the four of us that read and reviewed the paper feel less convinced and less supportive rather than more when such grand statements are made based on so little evidence.

We understand the reviewer’s concerns and examined these autoregulated TFs in LB and M9 glycerol media. Data from these conditions support our previous claim in line 152-153 (previous submission): (1) multiple autoregulated TFs described in previous databases (e.g., Fur, MazF, CreB, YafQ) did not show auto-regulatory response under any of the growth conditions tested in PPTP-seq; (2) some TFs only showed auto-regulatory response under specific conditions (e.g., LldR autoregulation was observed in LB media but not in M9 glucose medium; UvrY autoregulation was observed in M9 glycerol but not in M9 glucose; ComR autoregulation occurs in M9 glucose and M9 glycerol but not in LB) (Supplementary Figure 12e, below). To make it more clear, we revised this sentence to “some

previously identified TFs lack autoregulatory response when cells are growing in minimal glucose medium and may occur under other growth conditions¹⁰⁻¹³, so the interpretation of TF regulation should consider the condition dependency”.

Supplementary Figure 12e Comparison of auto-regulatory responses measured by PPTP-seq in minimal glucose, LB, and minimal glycerol media.

- Line 193: We have no idea what “relatively stronger local binding strength” – relative to what? What is this about?

When multiple TFs binding to the same promoter, we compared their relative binding strengths and found that TFs strongly bound to the promoter are more likely to affect promoter activity. To clarify it, we changed the subtitle to “Strongly bound rather than weakly bound TFs tend to affect promoter activity”.

- Line 209-211: “48% of promoters were regulated by more than one TF”, but earlier you said that 31% (line 117) were regulated by one activator or one repressor. On top of that, 62% of all promoters were variable. These numbers do not add up, please clarify.

The numbers “31%” and “62%” are from this study in M9 glucose condition: 62% of all promoters were variable, among them, 31% of variable promoters were regulated by one activator or one repressor. The number “48%” is from RegulonDB. To avoid confusion, we revised our manuscript (lines 245-246).

- Line 215-217: Please refer to Rydenfelt, Mattias, et al. "The influence of promoter architectures and regulatory motifs on gene expression in Escherichia coli." PLoS one 9.12 (2014): e114347.

We added this reference in the revised manuscript.

- Line 218-228: please consider referring to Garcia & Phillips, 2011; Razo-Mejia et al., 2018, Cell Systems, Belliveau et al., 2018, PNAS; Barnes et al., 2019, PLOS Comp Bio

We added these references in the revised manuscript.

- Line 239-241: Refer to other growth conditions. Although work in this study does uncover information about individual TFs, these findings do not qualify as “unknown regulatory rules”.

We changed the term from “unknown regulatory rules” to “unknown regulatory responses”.

- Line 250-251: This is no different to other methods such as deep mutational scanning

Deep mutational scanning is to perturb cis-regulatory elements (e.g., promoters) in gene regulation while PPTP-seq is to perturb trans-regulatory elements (e.g., TFs). Deep mutational scanning alone without prior knowledge of TF binding sites cannot provide information about which TF a promoter can respond to. This information can be provided by PPTP-seq.

- Line 273-276: What is strong evidence? And why were 183 picked, when 181 had “strong evidence”?

The term “strong evidence” comes from RegulonDB, where it is described as “Single evidence with direct physical interaction or solid genetic evidence with a low probability for alternative explanations” (<https://regulondb.ccg.unam.mx/evidenceclassification>). A TF can be encoded by either one gene or multiple genes. The 181 TFs defined by RegulonDB v 10.0 are encoded by 183 unique genes. In this work, the sgRNA library targets 183 unique TF genes.

We further clarify our description in the Methods:

“From the existing TF-gene network (RegulonDB v 10.0), 181 TFs (including heteromultimeric TFs) were identified in *E. coli* that have at least one known target supported by binding of purified proteins or site mutation. Among them, 169 TFs function as monomers or homo-oligomers (encoded by single genes), and 12 TFs function as hetero-dimers or hetero-oligomers (encoded by more than one gene). The 181 TFs are encoded by 183 unique genes, thus a sgRNA library targeting 183 different TF genes were designed. (Supplementary Table 10).”

- Discussion: Discuss possible complementary uses with other methods

In revised manuscript, we added discussion on possible complementary uses with other methods (line 297-303). We further discussed potential applications and limitations of PPTP-seq in lines 312-329.

- Figure 1d: It is impossible to see the gray dots outside of the 2-fold change line, which we think are supposed to be the most significant ones?

We have revised Fig 1d to make dots outside 2-fold change bigger. A few differentially expressed variants are highlighted in Figure 2a. In Figure 1d, we want to emphasize that most promoter activities (>98%) are within 2-fold change.

- Figure 2a: Isn't this the same data as in Figure 1d, but replotted? Figure 1d,

Figure 1d shows the promoter activity level of all variants. Each row in y-axis represents a promoter. In figure 2a, the y-axis is the adjusted statistical significance and x-axis is the fold change relative to negative control. Both x-axis and y-axis are different in these two figures.

- Figure 2 b-c: Barplot on a log scale is very hard to interpret. The x-axis labels are backwards and should be “the number of promoters *regulating* a specific gene” and “the number of TF genes *regulated by* a specific promoter.

The log scale was changed to linear scale in the revised manuscript. The original labels are not backwards. To avoid confusion, we simplified the labels to “the number of regulated promoters per TF” and “The number of regulating TFs per promoter”

- Generally, the figures are just a big disappointment. For example, how do these authors expect people like us that REALLY care about the nature of the regulatory linkages in E. coli and other bacteria, to truly learn something from Fig. 3a? It feels as though no attempt was made to be creative or pedagogical or helpful to people like Collado that do RegulonDB to make figures that help us understand in actionable terms what is learned about each gene. Sure, it is a lot of work, but for example, maybe they need in the SI or a website to take EVERY gene they looked at and make a cartoon that SIMPLY summarizes what they think happened with that gene. For example, earlier in the paper, they made a claim about simple repression and simple activation being the most common motifs. Great, so why not make a few tables that say these are ALL the genes we found are regulated by simple activation and here is the name of the TF that does the activation? Here are ALL the genes we found that are regulated by simple repression and here is the name of the TF that does the repression?

We thank the reviewer for the comments. We extensively revised Fig 3a to incorporate names of important TF-target pairs. We believe the current version shows perturbation-response relationship between TFs more clearly. We further added a table (Supplementary Table 5) to summarize these results. We also added another table (Supplementary Table 6) to compare binding data sets with promoter responses we measured for all three growth conditions.

Though we fear the editors will take our remarks as the basis for rejection, that is NOT what we are arguing. This is a very interesting approach, and seems to have led to some insights. But, for slow thinkers like us, why can't these authors work harder to not fall into the typical trap of modern genomic science with figures that are largely useless and instead really trying to mine their gold mine of data in ways that make it accessible in a practical way?

We believe the additional experiments performed in different growth conditions, revised figures, Supplementary Tables, Notes, and explanations in the revised submission have demonstrated the PPTP-seq results more clearly. The additional regulatory responses from new growth conditions should address many of the reviewer's concerns.

Reviewer #2

In the present manuscript, Han Y. et al developed a new technique called pooled promoter responses to TF perturbation sequencing (PPTP-seq) to measure the activity of 1373 E. coli promoters under single knockdown of 183 TFs via CRISPR interference and illustrated more than 200,000 possible TF-gene responses in one experiment. The PPTP-seq enabled the authors to identify novel TF auto-regulatory responses and also novel transcriptional control in one-carbon metabolism. Finally, by comparing their data to the published TF binding peaks identified by in vitro DAP-seq (PMID: 34824476), the authors found that relative binding strengths of TFs on a promoter were a major determinant of promoter response. In general, this study showed the power of the PPTP-seq in the study of high throughput TF-gene responses in E.coli. However, several concerns need to be clarified.

We thank the reviewer for the overall positive evaluation of our work.

1. It is important to know the repression efficiency of CRISPR interference on the expression of selected TFs. However, this paper lacks such knowledge.

We thank the reviewer for raising this concern. We have used two approaches to evaluate the repression efficiency of CRISPRi in our library. First, we examined sequences in the promoter library that can be directly targeted by sgRNAs, thus the CRISPRi repression efficiency of a TF can be estimated by the fold of repression on its own promoter. Totally 41 TFs have their promoters in our library, and 39 of them (95%) showed repression (Student's t-test P value < 0.05) (Supplementary Fig 5a). Second, additional RT-qPCR experiments were performed to examine the CRISPRi repression efficiency. The nine TFs that were used to examine individual responses (Supplementary Fig 6) and another five TFs (HipB, NarP, OmpR, YefM, and YeoB) randomly selected were chosen as representatives. The RT-qPCR results showed clear repression for 12 out of 14 TFs, representing 86% is effective (Supplementary Fig 5b). We further found a clear negative correlation between the degree of CRISPRi repression and TF expression level measured from TF promoter activity (Supplementary Fig 5c, d). This explains the lack of repression for the small fraction of TFs (e.g., *qseB* and *ttdR*).

We updated these results in main text (lines 105-112) and Supplementary Note 1.

Supplementary Figure 5. Evaluation of CRISPRi repression efficiency.

a and **b** Repression of TF genes by CRISPRi measured by PPTP-seq (**a**) and RT-qPCR (**b**). Data points show biological replicates. **c** Relative TF expression level as measured by their promoter activities in PPTP-seq using the strain with non-targeting control sgRNA. **d** Relationship between CRISPRi repression activity and TF promoter activity without CRISPRi repression.

2. In line 75-76, transformed cells were grown in minimal glucose medium to a steady state and sorted into 16 bins based on their fluorescence intensity (Supplementary Fig. 1a). The authors should give reasons that why sort the cells into 16 bins (why not more bins) and describe how they quantify the expression in the main text in more details. It is difficult to understand how they did the quantification as described in the method. Also it is necessary to give the fold changes (non-targeting sgRNA vs a sgRNA targeting) in Supp. Fig 2. In Fig. 2a, most of the Log2 fold changes of the varied TF-gene pairs were smaller than 4. Especially, the fold change of the varied LacI-LacZ pairs was much smaller than the reported number. The resolution of the expression quantification of the promoter could be quite low and the rather low resolution of the expression quantification may hide more varied TF-gene pairs.

We thank the reviewer for pointing out these issues. Here we address them individually.

1. *Why 16 bins*: The number of bins was chosen by considering both sorting time and gene expression resolution (less than 1.7-fold change within a bin). The effects of bin numbers to gene expression quantification were studied in our previous sort-seq paper as well as by other groups⁵⁻⁹. All these

studies have shown that mean gene expression can be reliably measured using 1.7-fold change (equivalent to 16 bins in our gene expression scale). More bins with the same number of sorted cells will result in longer sorting time. We added further explanation on the choice of 16 bins in the revised manuscript (lines 415-418)

2. *Quantification method*: To estimate promoter activities under each perturbed TF condition, sequencing read counts across the bins were first converted to cell count distribution for each individual variant, followed by fitting into log-normal distribution by maximum-likelihood estimation^{8,9,14} (Supplementary Fig. 2 and Methods). We added this detail to the main text (lines 86-89).

3. *Fold changes in Supplementary Fig. 2*: We added the fold changes to Supplementary Fig. 2.

4. *Fold changes of *LacI-lacZ* pair*: Our fold change quantification of *lacZ* promoter activity is different from that of other studies due to a few differences. We believe a major difference is the lack of DNA looping. The native *lacZ* gene contains an operator site, O2, in the *lacZ* coding region that generates a DNA loop with the O1 operator site, whereas the *lacZ* promoter in our reporter plasmid does not have the additional operator site thus cannot form DNA looping. DNA loop has been shown to result in additional >10-fold repression¹⁵. Another difference is the DNA copy number of the promoter. Our reporter system contains 3-5 additional copies of the *lacZ* promoter in the reporter plasmid. Previous TF titration studies have shown that varying the ratio of LacI to its operator sites can create 10- to 50-fold change on promoter activity¹⁶. To clarify these points, we discussed these explanations in the revised manuscript in lines 325-329.

3. In line 85-87, out of 182 tested TFs, 178 of them were able to activate or repress at least one promoter. The author should compare their data to the published data set to validate their findings. In Fig 2d, it is also necessary to compare the variable and constant promoters identified here to the published data.

We thank the reviewer for the suggestion of validating our finding using published data. In the revised manuscript, we added PPTP-seq from two additional growth conditions (minimal glycerol and LB), and we compared PPTP-seq data from all three conditions to published data. Out of 576 direct regulatory interactions in RegulonDB, 115 TF-promoter pairs showed promoter activity change by TFKD in PPTP-seq in at least one condition (Supplementary Table 8). We also compared our data with expression profiles from RNA-seq, microarray, and flow cytometry (Supplementary Figure 7), and with binding datasets from ChIP-seq, gSELEX, and DAP-seq (Supplementary Table 6).

Variable and constant promoters are defined as promoters with significant activity change under at least one TFKD condition and promoters without any significant activity change under all TFKD conditions, respectively, by our paper, thus are not compared to previous works.

4. In Fig 3a, the regulatory network looks very complicated, and this panel was even not cited in the main text. The authors may find a way to highlight the most important one and also compare this finding to the published data.

We thank you for this suggestion. Fig 3a has been reorganized by grouping different TFs to make the regulatory network more clearly. We further compared our response data with previous published binding data sets (Fig. 3a and Supplementary Table 5). This panel was also cited in the revised manuscript.

5. In line 176-177, these previously unidentified MetR regulations were further verified using the tunable TF library (Fig. 4c). However, this was done in the MetR knockdown background. Are the newly identified MetR targets also identified by the reported *in vitro* DAP-seq?

Thank you for pointing this out. The MetR knockdown strain was from the tunable TF library. To avoid confusion, we revised the sentence to “in a MetR knockdown strain” to be consistent with the figure legend.

We further checked the *in vitro* DAP-seq data. MetR binding sites were identified on *metF* and *folE* promoters, but not in *thyA* promoter. We revised the manuscript to reflect this finding (lines 205-207).

6. How much can we trust the results reported in Fig 5c-f, since only the regulatory or non-regulatory information were obtained in this work *in vivo*, while the other important parameters including TF concentration and binding strength were obtained by other labs *in vitro* or in different conditions?

We understand the reviewer’s concern. The TF copy numbers (used in Fig 5e) were also determined experimentally in minimal glucose condition, although by other groups¹⁷. The relative binding strengths (Figs 5g, i) were calculated from theoretical binding capacity at an optimal condition, which can be different from our *in vivo* test condition. Discrepancies between our *in vivo* work and other *in vitro* work indeed exist for individual TF-promoter pairs. However, the overall dataset from all interactions may suggest features that broadly affect regulation. In our revised manuscript, we also include TF copy numbers measured by mass spectrometry (Fig. 5f) and relative binding strength per TF measured by gSELEX (Fig. 5h), and the results are similar (Fig. 5e-h).

Reviewer #3 (Remarks to the Author):

In their manuscript “Genome-wide promoter responses to CRISPR perturbations of regulators reveal regulatory networks in *Escherichia coli*” Han et al. describe PPTP-seq, a pooled high-throughput method of evaluating promoter activity in the presence of CRISPRi knockdowns of transcription factors. The authors work represents a major advance. However, additional work needs to be done characterize the performance of PPTP-seq and to better connect it to previous methods. I outline these issues in the text below:

We thank the reviewer for the overall positive evaluation of our work and the additional work mentioned to strengthen this manuscript.

Major points.

1. PPTP-seq fundamentally provides the same kind of information as ChIP-seq, DAP-seq, SELEX, RNA-seq, and many other methods. Therefore, it is important to provide a more through comparison of its plusses and minuses of their method compared to the others. Though the authors provide a comparison with DAP-seq, this is not enough, especially given how well characterized many *E. coli* TFs are.

We thank the reviewer for this suggestion. In the revised manuscript, we provide a deeper comparison between PPTP-seq with ChIP-seq, ChIP-exo, DAP-seq, gSELEX, RNA-seq, microarray, and another promoter library. Please see lines 123-129, lines 224-233, Supplementary Figure 7 (comparison with RNA-seq, flow cytometry, and EcoMAC microarray), Figure 5 and Supplementary Table 6 (comparison with DAP-seq, gSELEX, ChIP-seq, ChIP-exo, and curated TFBSs in RegulonDB) for details.

a. After reading the manuscript, I have no idea what the expected false positive and false negative rates are, nor the distribution between direct and indirect regulatory interactions among the results.

We thank the reviewer for pointing out the discussion of expected false positive and false negative, which we think are important and now added to the revised manuscript (lines 320-325 and Supplementary Note 5).

PPTP-seq is based on gene response to TF perturbation, thus false positives can be caused by off-target and polarity of CRISPRi (i.e., response to unexpected perturbations¹⁸⁻²¹). Previous genome-scale CRISPRi screens in *E. coli* showed an off-target probability of 10.7%¹⁸ and a polarity probability is 8.0%¹⁹. Our sgRNA design eliminated bad-seed effects¹⁸ that cause off-target in the previous CRISPRi screens, thus is expected to have reduced off-target rate (<10.7%). Additionally, if the unexpectedly perturbed gene was not a TF gene, it is less likely to cause any promoter activity changes. Considering these, the expected false positive rate should be substantially less than $1-(1-10.7%)*(1-8%)=17.8%$, if we treat off-target and polar effects independently, but the real false positive rate should be much lower than this number.

False negatives can arise from inefficient TF repression (i.e., not creating an effective perturbation). As we discussed in comment 3b below, CRISPRi repression tested in minimal glucose

condition showed that 4 out of 52 TFs were not significantly repressed by CRISPRi, which gives an expected false negative rate about 8%, if we equally treat every TF.

Discussions on direct v.s. indirect regulatory interactions are another good point. PPTP-seq only reveals differences in gene expression under TF perturbation, thus in theory cannot differentiate direct v.s. indirect regulatory interactions. However, if previous binding information can be used as evidence for direct regulation, an estimation can be made. In minimal glucose condition, Fig 5a and 5b showed that the PPTP-seq data set may have approximately 12% direct interactions. This discussion is added to lines 224-233 and 312-319.

b. How do your steady state promoter activity measurements compare to RNA-seq and/or nascent RNA-seq measurements of the same?

In the revised manuscript, we compared PPTP-seq measurements in negative control strains to previous RNA-seq data²². The Pearson's correlation coefficient between $\log_{10}(\text{GFP})$ from PPTP-seq and $\log_{10}(\text{transcript per million})$ from RNA-seq was 0.68 (Supplementary Fig. 7a). We further compared our dataset to promoter activity measurements from Alon's library measured by flow cytometry, showing a good correlation of 0.74 (Supplementary Fig. 7b). In fact, the correlation between PPTP-seq with RNA-seq (0.68) is better than the correlation between Alon's promoter activities with RNA-seq ($r = 0.58$) (Supplementary Fig. 7c). We believe the improved correlation in PPTP-seq is caused by the use of a self-cleaving ribozyme RiboJ placed between the promoter library and the *gfp* gene, eliminating interference of different promoter sequences with *gfp* mRNA stability. The improved correlation is particularly apparent for weak promoters (Supplementary Fig. 7c). Further, transcript abundance measured from RNA-seq depends on not only promoter activity, but also other factors, such as mRNA stability and processing, which can contribute to variations between RNA-seq and promoter activities measurements.

Additionally, fold change in promoter activity upon TFKD measured from PPTP-seq is also qualitatively consistent with that measured from EcoMAC microarray²³ (data source: https://github.com/CovertLab/wcEcoli/blob/master/reconstruction/ecoli/flat/fold_changes.tsv) (Pearson's $r = 0.51$, Supplementary Fig. 7d).

These results were added to main text in the revised manuscript (lines 123-129).

Supplementary Figure 7. Correlation of PPTP-seq data with published data sets.

a and **b** Promoter activity measured from PPTP-seq were compared to transcript level measured from RNA-seq dataset in Macklin et al.²² (**a**) and promoter activity measured by flow cytometry in Silander et al.²⁴ (**b**). **c** As a reference, data in Macklin et al. were compared to data in Silander et al. Correlation in **c** is weaker than correlations shown in **a** and **b**. **d** Comparison of fold changes at the log₂ scale (log₂FC) measured from PPTP-seq to log₂FC obtained from EcoMAC microarray²³. Red line represents a fitting result of linear regression. For each plot, number of values (n) and Pearson's correlation coefficient (r) are displayed. TPM, transcript per million. Source data are provided as a Source Data file.

2. Sequence analysis of promoters would be useful for understanding the nature of observed regulatory interactions. How often is it the case that a TF knockdown affects a promoter, but no binding site for that TF is apparent at that promoter?

To characterize TF binding sites at each promoter, we collected all experimental binding data sets in *E. coli*, including those from ChIP-seq, ChIP-exo, DAP-seq, gSELEX, and many manually curated TF binding sites in RegulonDB. Out of the 4058 regulatory responses identified by PPTP-seq in minimal glucose medium, 225 have binding evidence from DAP-seq and additional 256 have binding evidence from other binding datasets, representing for 12% of the PPTP-seq responses (Fig. 5a, b, Supplementary Table 6). For 127 TFs with binding site information, on average 23% of regulated

promoters per TF were presumably direct targets (Fig. 5c). For the rest 56 TFs, their TFBSs were either not in our promoter library or not identified yet.

These results were shown in the revised Fig 5a, b, and Supplementary Table 6, and were discussed in lines 224-233.

3. Several aspects of the paper are insufficiently described. For example:

a. A strong point of the paper is that the authors used the Alon collection of promoters. However, some basic summary about how these promoters were chosen and how much of the sequence they encompass is really important for interpreting the authors' results and shouldn't require readers to find the Alon paper. Listing all of the promoters in a supplementary table would also be helpful.

We thank the reviewer for these suggestions. The information has been added to the revised manuscript in lines 354-358 and 440-443. In brief, all promoters in the Alon's collection were included in our PPTP-seq. In Alon's collection, "promoter regions" were defined as entire intergenic regions flanked by about 50-150 bp into the adjacent coding regions¹. In our experiment, we excluded promoters inside operons according to RegulonDB v10.0 operon structure annotation. Following reviewer's suggestion, promoters used in PPTP-seq were listed in Supplementary Table 2.

b. What CRISPRi system did the authors use? How was it induced, and what was the approximate knockdown of the TFs?

We used the CRISPRi system developed by David Bikard lab^{18,25}. The sgRNA is controlled by a strong constitutive promoter J23119. The *dCas9* gene was integrated into the chromosome and was placed under the control of a tight aTc-inducible promoter P_{tet}. Thus, CRISPRi was induced by adding aTc, and cells were maintained under steady state growth after induction before data collection. The details can be found in Methods lines 342-344 and 402-409.

We thank the reviewer for raising this concern. We have used two approaches to evaluate the repression efficiency of CRISPRi in our library. First, we examined sequences in the promoter library that can be directly targeted by sgRNAs, thus the CRISPRi repression efficiency of a TF can be estimated by the fold of repression on its own promoter. Totally 41 TFs have their promoters in our library, and 39 of them (95%) showed repression (Student's t-test P value < 0.05) (Supplementary Fig 5a). Second, additional RT-qPCR experiments were performed to examine the CRISPRi repression efficiency. The nine TFs that were used to examine individual responses (Supplementary Fig 6) and another five TFs (HipB, NarP, OmpR, YefM, and YeoB) randomly selected were chosen as representatives. The RT-qPCR results showed clear repression for 12 out of 14 TFs, representing 86% is effective (Supplementary Fig 5b). We further found a clear negative correlation between the degree of CRISPRi repression and TF expression level measured from TF promoter activity (Supplementary Fig 5c, d). This explains the lack of repression for the small fraction of TFs (e.g., *qseB* and *ttdR*).

Supplementary Figure 5. Evaluation of CRISPRi repression efficiency.

a and **b** Repression of TF genes by CRISPRi measured by PPTP-seq (**a**) and RT-qPCR (**b**). Data points show biological replicates. **c** Relative TF expression level as measured by their promoter activities in PPTP-seq using the strain with non-targeting control sgRNA. **d** Relationship between CRISPRi repression activity with TF promoter activity without CRISPRi repression.

We updated these results in main text (lines 105-112) and Supplementary Note 1.

c. Since DAP-seq plays an important role in the paper, perhaps a few words about what it is may help readers understand what's going on.

We briefly described DAP-seq in our revised manuscript (lines 205-206)

4. The authors should address the issue of CRISPRi polarity, especially since bacterial TFs are frequently located near the genes they regulate. For example, marR is located immediately upstream of its targets marA and marB. Similarly, nagC is located in the middle of its operon, and mraZ is located at the start of the (highly essential) dcw operon.

Thank you for pointing this out. We did observe CRISPRi polarity issue in our experiments. However, the chance for CRISPRi polarity to affect our perturbation response results is low, because perturbation of TF genes is much more likely to affect promoter activity compared to perturbation of non-TF genes²³. If more than one TF are located within the same operon, it may have substantial influence. However, only 22 TFs are encoded within one multi-TF operon.

In our revised manuscript, we analyzed the possibilities of all TF genes for having CRISPRi polarity and added the results to Supplementary Table 1. In general, 91 TF genes locate in a single gene operon, which are unlikely to involve CRISPRi polarity. 45 TF genes are the first gene of a multi-gene operon, which may only cause forward polarity for downstream genes. 34 TF genes are the last gene of a multi-gene operon, which may only cause reverse polarity for upstream genes. The rest 13 TF genes in the middle of a multi-gene operon may cause both forward and reverse polarity. In the Supplementary Table 1, we highlighted operons with multiple TFs that could generate alternative explanations in our PPTP-seq results.

In the revised discussion section, we also discussed how CRISPRi polarity may causes false positive as a limitation to PPTP-seq and other CRISPRi based technology in bacteria.

We further discussed how cell growth may be affected by CRISPRi polarity in lines 102-104.

5. Most sequencing based assays compare the relative abundance of different strains. In contrast, this analysis relies on accurately quantifying absolute abundance. Quantifying absolute abundance allow direct comparisons between bins. However, in order for the method to be valid, the authors need to demonstrate that the same number reads reflect the same number of cells in each bin. It is not clear to me how the authors assure this through the different plasmid extraction and library prep steps. A simple method for normalizing reads would be to sequence an unsorted (but otherwise identically treated) sample of the library and then apply a linear regression to determine relative weights for each of the bins. ($\text{unsorted_reads} = w_1 \cdot \text{bin}_1 + w_2 \cdot \text{bin}_2 \dots \text{etc}$).

We did not directly use the number of reads to reconstruct GFP distribution for each variant, but convert the number of reads in each bin to relative number of cells. During sorting, we have ensured that the number of cells we collected in each bin matches the expected number of cells belonging to that bin in unsorted population. The expected number of cells of a variant i in bin j (c_{ij}) is the total number of cells in bin j (C_j) times the ratio of variant i in bin j (ratio_{ij}). This ratio was measured by sequencing. We added more details to clarify our method.

“For each variant i , its read counts r_{ij} in each bin j were multiplied by C_j/R_j to estimate its cell counts c_{ij} sorted to bin j ,

$$c_{ij} = C_j \times \frac{r_{ij}}{R_j} = r_{ij} \times C_j/R_j$$

where C_j and R_j are the number of cells collected in bin j and the number of reads sequenced with barcode associated with bin j respectively. This normalization step allows comparisons between bins after post-sort growth, plasmid extraction, and NGS preparation by assuming the ratio of each variant in a bin does not change significantly.”

Minor points

1. L19 – you are not measuring “phenotypic responses”, but gene expression.

The phrase “phenotypic responses” was changed to “gene expression responses” in the abstract.

2. L32 – “Information about bacterial cellular responses is mostly encoded in promoters...” This is an overly broad statement with no supporting information that serves only to antagonize people who study non-transcriptional regulation. Plenty of bacterial responses involve primarily non-transcriptional responses.

These sentences were revised as “Information about the bacterial cellular response is often encoded in promoters and affected by transcription factors (TFs), which control both the timing and level of gene expression.”

3. L46-L47 – “this method...cannot provide information about which TF is regulating the promoter.” Neither can PPTP-seq, since knockdown of a TF can cause compensatory responses that involve other TFs.

In our revised manuscript, we changed it to “While powerful, this method without prior knowledge of TF binding sites alone cannot provide information about which TF the promoter can respond to.”

4. L57-59 – “Our study uncovered novel regulatory rules, including TF autoregulatory responses, complex transcriptional control of metabolic pathways, and promoter responses to coregulation.” You did not discover these “rules”, you discovered some (minor) examples that further reinforce these well-established principles.

We changed the term “rules” to “responses”.

5. L120-121 – “COG analysis”. Consider also doing the analysis using GO terms and KEGG categories. You may find more descriptive/useful functional annotations.

Thank you for the suggestion. We performed GO and KEGG analyses using the DAVID online tool. The results were added to the revised main text (lines 147-154) and Supplementary Table 4. Enriched GO and KEGG functional categories were consistent with COG analysis and are more specific in terms of functional description.

6. L143 – PgrR is known to autoregulate. They even did a gel shift experiment.

<https://doi.org/10.1111/gtc.12026>

Thank you for pointing this out. PgrR was not recognized for autoregulation in the latest RegulonDB. We are happy to see that the cited paper showed evidence of self-regulation that is consistent with our observation. We revised our manuscript and cited this reference (line 176).

1. Zaslaver, A. *et al.* A comprehensive library of fluorescent transcriptional reporters for *Escherichia coli*. *Nat. Methods* **3**, 623–628 (2006).
2. Ireland, W. T. *et al.* Deciphering the regulatory genome of *Escherichia coli*, one hundred promoters at a time. *Elife* **9**, 1–76 (2020).
3. Belliveau, N. M. *et al.* Systematic approach for dissecting the molecular mechanisms of

- transcriptional regulation in bacteria. *Proc. Natl. Acad. Sci. U.S.A.* **115**, E4796–E4805 (2018).
4. Urtecho, G. *et al.* Genome-wide functional characterization of Escherichia coli promoters and regulatory elements responsible for their function. *BioRxiv* (2020). doi:10.1101/2020.01.04.894907
 5. Schmitz, A. & Zhang, F. Massively parallel gene expression variation measurement of a synonymous codon library. *BMC Genomics* **22**, 1–12 (2021).
 6. Cambray, G., Guimaraes, J. C. & Arkin, A. P. Evaluation of 244,000 synthetic sequences reveals design principles to optimize translation in escherichia coli. *Nat. Biotechnol.* **36**, 1005 (2018).
 7. Sharon, E. *et al.* Inferring gene regulatory logic from high-throughput measurements of thousands of systematically designed promoters. *Nat. Biotechnol.* **30**, 521–530 (2012).
 8. Kotopka, B. J. & Smolke, C. D. Model-driven generation of artificial yeast promoters. *Nat. Commun.* **11**, 2113 (2020).
 9. Peterman, N. & Levine, E. Sort-seq under the hood: implications of design choices on large-scale characterization of sequence-function relations. *BMC Genomics* **17**, 1–17 (2016).
 10. Seo, S. W. *et al.* Deciphering fur transcriptional regulatory network highlights its complex role beyond iron metabolism in Escherichia coli. *Nat. Commun.* **5**, (2014).
 11. Marianovsky, I., Aizenman, E., Engelberg-Kulka, H. & Glaser, G. The regulation of the Escherichia coli mazEF promoter involves an unusual alternating palindrome. *J. Biol. Chem.* **276**, 5975–84 (2001).
 12. Gao, R. & Stock, A. M. Evolutionary tuning of protein expression levels of a positively autoregulated two-component system. *PLoS Genet.* **9**, e1003927 (2013).
 13. Aguilera, L. *et al.* Dual role of LldR in regulation of the lldPRD operon, involved in L-lactate metabolism in Escherichia coli. *J. Bacteriol.* **190**, 2997–3005 (2008).
 14. Townshend, B., Kennedy, A. B., Xiang, J. S. & Smolke, C. D. High-throughput cellular RNA device engineering. **12**, (2015).
 15. Oehler, S., Amouyal, M., Kolkhof, P., Von Wilcken-Bergmann, B. & Müller-Hill, B. Quality and position of the three lac operators of E.coli define efficiency of repression. *EMBO J.* **13**, 3348–3355 (1994).
 16. Brewster, R. C. *et al.* The transcription factor titration effect dictates level of gene expression. *Cell* **156**, 1312–1323 (2014).
 17. Li, G. W., Burkhardt, D., Gross, C. & Weissman, J. S. Quantifying absolute protein synthesis rates reveals principles underlying allocation of cellular resources. *Cell* **157**, 624–635 (2014).
 18. Cui, L. *et al.* A CRISPRi screen in E. coli reveals sequence-specific toxicity of dCas9. *Nat. Commun.* **9**, 1–10 (2018).
 19. Wang, T. *et al.* Pooled CRISPR interference screening enables genome-scale functional genomics study in bacteria with superior performance-net. *Nat. Commun.* **9**, (2018).
 20. Peters, J. M. *et al.* A comprehensive, CRISPR-based functional analysis of essential genes in bacteria. *Cell* **165**, 1493–1506 (2016).

21. Peters, J. M. *et al.* Bacterial CRISPR: Accomplishments and prospects. *Curr. Opin. Microbiol.* **27**, 121–126 (2015).
22. Macklin, D. N. *et al.* Simultaneous cross-evaluation of heterogeneous E. coli datasets via mechanistic simulation. *Science* **369**, eaav3751 (2020).
23. Carrera, J. *et al.* An integrative, multi-scale, genome-wide model reveals the phenotypic landscape of E. coli. *Mol. Syst. Biol.* **10**, 735 (2014).
24. Silander, O. K. *et al.* A genome-wide analysis of promoter-mediated phenotypic noise in Escherichia coli. *PLoS Genet.* **8**, 836–845 (2012).
25. Rousset, F. *et al.* Genome-wide CRISPR-dCas9 screens in E. coli identify essential genes and phage host factors. *PLoS Genet.* **14**, 1–28 (2018).

Reviewers' Comments:

Reviewer #2:

Remarks to the Author:

I thank the authors very much for all the additional experiments and modifications which make the global method of PPTP-seq more transparent. However, I still have several questions.

1. Regarding the network, what is the new finding of this work after comparing to the reported literatures? Only the few genes as show in Figure 4 and described in L204-208? For example, for the 4058 regulatory responses identified by PPTP-seq in glucose, which were the new findings in this work?
2. In Figure 5, what will be the relative binding strength per promoter measured by gSELEX? Also similar as that measured by DAP-seq in Figure 5I?
3. In the added part of condition-specific regulatory networks, it is very surprising to see that Crp showed more regulation on lacZ and malK in LB than that in glucose since people have known that cAMP level in LB was much lower. This kind of result could easily be interpreted by experimental errors, which was not real condition-specific regulation. Hence, these errors could make some of the identified regulations questionable.

Reviewer #3:

Remarks to the Author:

The authors' revision has largely addressed our comments.

Reviewer #2 (Remarks to the Author):

I thank the authors very much for all the additional experiments and modifications which make the global method of PPTP-seq more transparent. However, I still have several questions.

1. Regarding the network, what is the new finding of this work after comparing to the reported literatures? Only the few genes as show in Figure 4 and described in L204-208? For example, for the 4058 regulatory responses identified by PPTP-seq in glucose, which were the new findings in this work?

In glucose, PPTP-seq identified 4058 regulatory responses, including 481 with binding evidence and 3577 without binding evidence. Out of the 481 responses with binding evidence, 78 were found in the TF-operon network in RegulonDB while 403 responses are new. For the 3577 responses without binding evidence, none were found in the TF-operon network in RegulonDB. These new responses were discussed in lines 233-236.

We also analyzed the new responses in LB and glycerol summarized these values in a new Supplementary Table 7.

Medium	Minimal Glucose		LB		Minimal Glycerol	
PPTP-seq responses	4058		5279		3810	
Binding evidence	With	Without	With	Without	With	Without
Total	481	3577	450	4829	373	3537
Known from RegulonDB	78	0	53	1	43	0
New response	403	3577	397	4828	230	3537

2. In Figure 5, what will be the relative binding strength per promoter measured by gSELEX? Also similar as that measured by DAP-seq in Figure 5I?

We thank the reviewer for the suggestion to examine the “relative binding strength per promoter” by gSELEX. Unfortunately, the gSELEX database (called Transcription Profile of E. coli, TEC in short)¹ does not allow us to calculate the “relative binding strength per promoter”. In the gSELEX paper, to adjust the background signal in binding, TF binding intensities have all been normalized to the highest binding intensity of each TF, but not to each promoter. Thus gSELEX can generate data for “relative binding strength per TF”, but not “relative binding strength per promoter”. Additionally, in the gSELEX experiment, the background binding signals for each TF may be different because they were collected using different probes and the number of SELEX cycles varies for each TF. Thus, we think we cannot use the gSELEX data to compare binding of different TFs to the same promoter.

3. In the added part of condition-specific regulatory networks, it is very surprising to see that Crp

showed more regulation on lacZ and malK in LB than that in glucose since people have known that cAMP level in LB was much lower. This kind of result could easily be interpreted by experimental errors, which was not real condition-specific regulation. Hence, these errors could make some of the identified regulations questionable.

We believe that our data is consistent with published literatures on cAMP-CRP regulation. cAMP level is inversely proportional to glucose level. High glucose sharply reduced the intracellular cAMP by an indirect mechanism: glucose decreases the level of phosphorylation of PTS, which is an activator of adenylate cyclase (*cyaA*) for cAMP synthesis². Since LB medium does not contain a high glucose concentration, cAMP level is expected to be higher in LB than that in M9 glucose. This idea is supported by the following references that measured cAMP concentrations in exponentially growing *E. coli* in M9 glucose and LB media:

In LB medium, cAMP concentration was reported to be 5.4 (± 1.06) pmol/OD/mL, which is equivalent to 0.72 (0.60 to 0.84) mM extracellular cAMP in the $\Delta cyaA$ mutant³. At this cAMP concentration, cAMP-CRP regulated hundreds of genes in the genome³.

In contrast, cAMP in M9 glucose medium was respectively reported to 0.035 (0.028 to 0.044) mM⁴ and 0.025 (0.019 to 0.031) mM by two different groups⁵. At this concentration, the activity of cAMP-CRP was nearly zero⁵.

Therefore, when CRISPRi decreased CRP expression in LB, the concentration of cAMP-CRP complex decreased, losing the activation of LacZ expression (Fig 6b). In contrast, cells have low cAMP concentration in M9 glucose medium, thus decreasing CRP expression had little effect to the concentration of cAMP-CRP complex, leading to little change in LacZ expression (Fig 6b).

Reviewer #3 (Remarks to the Author):

The authors' revision has largely addressed our comments.

We thank the reviewer for examining our revised manuscript.

References:

1. Ishihama, A., Shimada, T. & Yamazaki, Y. Transcription profile of Escherichia coli: Genomic SELEX search for regulatory targets of transcription factors. *Nucleic Acids Res.* **44**, 2058–2074 (2016).
2. Kolb, A., Busby, S., Buc, H., Garges, S. & Adhya, S. TRANSCRIPTIONAL REGULATION BY cAMP AND ITS RECEPTOR PROTEIN. *Annu. Rev. Biochem.* **62**, 749–797 (1993).
3. Chakraborty, S., Singh, P. & Seshasayee, A. S. N. Understanding the Genome-Wide Transcription Response To Various cAMP Levels in Bacteria Using Phenomenological Models. *mSystems* **7**, (2022).
4. Bennett, B. D. *et al.* Absolute metabolite concentrations and implied enzyme active site

occupancy in *Escherichia coli*. *Nat. Chem. Biol.* **5**, 593–599 (2009).

5. Lempp, M. *et al.* Systematic identification of metabolites controlling gene expression in *E. coli*. *Nat. Commun.* **10**, (2019).

Reviewers' Comments:

Reviewer #2:

Remarks to the Author:

I thank the authors for the revision.